



**Towards an improvement of OSL age uncertainties: modelling OSL ages with systematic errors,**
**stratigraphic constraints and radiocarbon ages using the R package 'BayLum'**
Guillaume Guérin[1,2*], Christelle Lahaye[1], Maryam Heydari[1], Martin Autzen[3,4], Jan-Pieter Buylaert[3,4],
Pierre Guibert[1], Mayank Jain[3], Sebastian Kreutzer[5,1], Andrew S. Murray[4], Kristina J. Thomsen[3], Petra
Urbanova[1], Anne Philippe[6].
*[1] UMR 5060 CNRS - Université Bordeaux Montaigne, IRAMAT-CRP2A, Maison de l'archéologie,*
*Esplanade des Antilles, 33607 Pessac cedex, France.*
*[2] Univ Rennes, CNRS, Géosciences Rennes, UMR 6118, 35000 Rennes, France*
*[3] Center for Nuclear Technologies, Technical University of Denmark, DTU Risø Campus, DK-4000*
*Roskilde, Denmark.*
*[4] Nordic Laboratory for Luminescence Dating, Department of Geoscience, Aarhus University, DTU Risø*
*Campus, DK-4000 Roskilde, Denmark.*
*[5]Geography & Earth Sciences, Aberystwyth University, Aberystwyth, Wales, United Kingdom*
*[6] Jean Leray Laboratory of Mathematics (LMJL), UMR6629 CNRS - Université de Nantes, France.*
**Keyword:**
OSL dating; Bayesian modelling; R package; Systematic errors; Covariance matrix; Stratigraphic
constraints
**Abstract**
Statistical analysis has become increasingly important in the field of OSL dating since it has become
possible to measure signals at the single grain scale. The accuracy of large chronological datasets can
benefit from the inclusion, in chronological modelling, of stratigraphic constraints and shared
systematic errors. Recently, a number of Bayesian models have been developed for OSL age
calculation; the R package 'BayLum' allows implementing different such models, in particular for
samples in stratigraphic order which share systematic errors. We first show how to introduce
stratigraphic constraints in 'BayLum'; then, we focus on the construction, based on measurement
uncertainties, of dose covariance matrices to account for systematic errors specific to OSL dating. The
nature (systematic versus random) of errors affecting OSL ages is discussed, based – as an example –
on the dose rate determination procedure at the IRAMAT-CRP2A laboratory (Bordeaux). The effects of
the stratigraphic constraints and dose covariance matrices are illustrated on example datasets. In
particular, the interest of combining the modelling of systematic errors with independent ages,
unaffected by these errors, is demonstrated. Finally, we discuss other common ways of estimating
dose rates and how they may be taken into account in the covariance matrix by other potential users
and laboratories. Test datasets are provided as supplementary material to the reader, together with
an R Markdown tutorial allowing to reproduce all calculations and figures presented in this study.



## 1. Introduction

Optically stimulated luminescence (OSL; Huntley *et al.*, 1985) allows dating the last exposure of quartz grains to sunlight. The Single Aliquot Regenerative (SAR) dose protocol consists of comparing the natural luminescence signal to laboratory-generated signals induced by artificial irradiations (Murray and Wintle, 2000; Wintle and Murray, 2006). The corresponding measurements, in particular at the single-grain scale, result in large datasets characterised by important scatter, owing to a number of dispersion factors (see, *e.g.* Thomsen *et al.*, 2005). An OSL age is then obtained by dividing the equivalent dose (i.e. in the case of coarse quartz grains, the dose absorbed by the mineral) by the dose rate to which quartz grains were exposed since the last exposure to light.

Statistical analysis, in the field of geochronology, generally aims at improving the precision, accuracy and/or range of dating methods. In the case of OSL dating, calibration errors on the laboratory source dose rate for natural dose estimation, and geochemical standards for dose rate assessment, have so far resulted in age uncertainties of, at best, ~5% (see, *e.g.*, Duller, 2008; Guérin *et al.*, 2013).

Note that in what follows, the unit of analysis is a sediment sample; the system of analysis is the laboratory in which the measurements are performed and includes both the apparatus and associated calibration standards. By definition, an error is the difference between the measured or observed value of a physical quantity, and its true (but unknown) value. Thus, by systematic errors, we refer to random errors affecting equipment calibration: whereas each of these errors may be assigned a Gaussian probability density function with zero mean and a known variance (the square root of the variance being generally referred to as uncertainty), at the scale of the laboratory this error takes a fixed, unknown value that affects all measurements in the same direction.

Over the past few years, several models for routine Bayesian analysis of SAR OSL and dose rate data were developed to reflect better, and take advantage of, the measurement procedures implemented to calculate OSL ages. Among those models, Combès *et al.* (2015) proposed one for calculating the central dose values for well-bleached samples, leading to higher overall accuracy (see Guérin *et al.*, 2015a) compared to the most commonly used model for OSL data analysis (the Central Dose Model: CDM, Galbraith *et al.*, 1999; note: we changed the original terminology following Galbraith and Roberts, 2012). Combès and Philippe (2017) developed models capable of dealing with individual and systematic multiplicative errors for OSL age calculation including stratigraphic constraints (for general introductions on a statistical analysis of OSL data, but also the statistical models discussed hereafter and associated prior distributions, the reader is referred to Combès *et al.*, 2015; Combès and Philippe, 2017, and references therein).

To implement the Bayesian models of Combès *et al.* (2015) and Combès and Philippe (2017) in practice, and provide easy access to the community, an R package (R Core Team, 2020) named 'BayLum' (Christophe *et al.*, 2020; version 0.2.0) has been developed and released on the Comprehensive R Archive Network (CRAN; see also Mercier *et al.*, 2017, for a first implementation of the central dose model from Combès *et al.*, 2015). First features of this 'BayLum' package were presented by Philippe *et al.* (2019) and its performances, when one is confronted either with large dose values or with dose variability issues, were tested in laboratory-controlled experiments (Heydari and Guérin, 2018) and later applied to various case studies (Lahaye *et al.*, 2018; Carter *et al.*, 2019; Heydari *et al.*, 2020; submitted; Chevrier *et al.*, accepted).

The purpose of this paper is to focus on the treatment of stratigraphic constraints and systematic errors for chronological modelling using 'BayLum', *i.e.* it goes beyond than what was first demonstrated by Philippe *et al.* (2019); together with the association of independent, more precise ages ([14]C in this work), such modelling is expected to reduce OSL age uncertainties. In the past, other





approaches to model systematic and random, individual errors in the field of palaeodosimetric dating
methods were proposed; in particular, Millard (2006a, 2006b) developed a Bayesian approach quite
close to that presented here, but which – among different other things (see Combès and Philippe,
2017, for a more detailed discussion) – is limited in its applicability.
Herein we present a Bayesian modelling case study. (1) We start with how data should be pre-
treated prior to using the 'BayLum' package; a simple example of chronological modelling (samples
considered independent, *i.e.* without stratigraphic constraints and shared errors) is first presented,
yielding an output from the 'BayLum' package to serve as a reference for the following, more elaborate
models. (2) In the next step, we detail how the user can integrate stratigraphic constraints and the
effect on the chronological inference. (3) Then, most importantly we explain how to build a dose
covariance matrix in practice to take into account systematic errors (for the definition of this matrix,
the reader is referred to Combès and Philippe, 2017) and what effect it has on a series of ages. (4) For
this purpose, we base our approach on dose rate measurements as performed by Guérin *et al.* (2015b)
at the IRAMAT-CRP2A laboratory. The effect of integrating independent data such as radiocarbon ages,
which do not share the systematic errors affecting OSL data, is then illustrated. (5) Finally, we discuss
different ways to measure dose rates and various assumptions that can be made regarding the nature
(systematic or random) of additional sources of errors in OSL dating.
To help the reader, we provide as supplementary information an R markdown document with
commented lines of code and example datasets, so that everything presented here may be
reproduced.
**2. Samples and methods**
**2.1. Case study**
To illustrate how to model OSL ages, both in stratigraphic constraints and sharing systematic
errors, using the R 'BayLum' package, we use the data from two sediment samples (FER 1 and FER 3)
already dated by quartz OSL (Guérin *et al.,* 2015b). These samples were taken from the archaeological
site of La Ferrassie (France) and prepared following standard chemical preparation procedures applied
to luminescence-dating samples. While modelling with 'BayLum' may be applied to both multi-grain
and single-grain OSL datasets, in the following we only focus on single-grain data, as this is probably
where the need for appropriate statistical models is most acute (the reliability of multi-grain OSL has
been demonstrated when using a plain average (mean) for palaeodose estimation; see, *e.g.,* Murray
and Olley, 2002; for theoretical justification, see Guérin *et al.,* 2017). Single-grain OSL data were
measured using an automated Risø TL/OSL reader (DA 20) fitted with a single grain attachment (Duller
*et al.*, 1999; Bøtter-Jensen *et al.*, 2000). A standard SAR protocol (Murray and Wintle, 2000; 2003) was
used to measure single-grain equivalent doses, after checking its suitability for the samples under
investigation. A comparison between quartz OSL and feldspar IRSL signals for these two samples, as
well as comparison with radiocarbon, showed that these samples were well-bleached at the time of
deposition and unaffected by post-depositional mixing. As a result, the use of central dose models is
fully justified (it should be noted here that at the time of writing, 'BayLum' does not yet include the
Bayesian model of Christophe *et al.,* 2018, allowing the analysis of poorly bleached samples).
**2.2. Data pre-treatment**
The Bayesian modelling implemented in 'BayLum' requires information of different natures: (i)
raw OSL data in the form of BIN/BINX file(s), (ii) list(s) of grains to be included in the modelling (based
on pre-defined selection criteria, *e.g.* on recycling and/or recuperation ratios), (iii) file(s) indicating how
the data should be processed (signal integration channels, reproducibility of the instrument(s), etc.)





and (iv) both natural (in Gy.ka$^{-1}$) and laboratory (in Gy.s$^{-1}$) dose rates. Based on these data, the
calculations are performed all at once using Markov Chain Monte Carlo (MCMC) computations; as a
result, unlike in standard frequentist data processing, there is no succession of steps in data analysis
(for example, individual equivalent dose estimates are not parameterised, unlike when the CDM is
used). While Combès *et al.* (2015) argue that this results in a better statistical inference about the age
(or palaeodose), it also comes with a downside: the user cannot visualise the data during the statistical
analysis. In particular, the fact that the user must specify the list of grains to be included in the analysis
implies that one should always pre-treat the samples in a standard way, by using, *e.g. Analyst* (Duller,
2015) or the R 'Luminescence' package (Kreutzer *et al.*, 2012; Kreutzer *et al.*, 2020) to visually check
the data but also investigate the effect of various selection criteria on the datasets (see for example
Thomsen *et al.*, 2016, on the effect of applying various selection criteria when with frequentist
statistical models; see Heydari and Guérin, 2018, for a similar study in a Bayesian framework).
In other words, using 'BayLum' for age calculation should not, and does not, prevent the user
from a careful and critical examination of the measured OSL data. In particular, before running age
calculations using the 'BayLum' package, it is important that the user already has identified potential
problems – *e.g.*, saturation and/or dose rate variability (see Heydari and Guérin, 2018, for adapted
modelling solutions).

**3. First simple model and output**

We first ran the function `Generate_DataFile()` for the OSL samples FER 1 and FER 3,
with the same lists of grains as those used for age calculation by Guérin *et al.* (2015b): all grains with
an uncertainty smaller than 20% on the first test dose signal were selected. A large number of grains
appeared to be in saturation for these samples (in *Analyst*, there is no intersection of the natural L/T
signal, or the sum of this sensitivity corrected natural signal and its uncertainty, with the dose-response
curve). As a result, following Thomsen *et al.* (2016) an additional selection criterion was added, based
on the curvature parameter of the dose-response curves. All grains for which the $D_0$ value, obtained
with Analyst as described by Guérin *et al.* (2015b), was smaller than 100 Gy, were rejected from the
analysis (note however that such a selection criterion may not be necessary when working with
'BayLum': Heydari and Guérin, 2018).
In practice, the data is contained in two folders named after the samples and provided as
Supplementary Material. Each folder contains one BIN/BINX-file (*i.e.* OSL measurements; note that
only a small fraction of the measured grains is included here Supplementary Material) and four CSV-
files:
- 'DiscPos.csv' lists all selected grains;
- 'Rule.csv' gives the rules for generating $L_x/T_x$ data (integration channels for both the natural
or regenerated and test dose signals, uncertainty arising from the reproducibility of the OSL
measurements, and number of SAR cycles to remove for curve fitting, if any - it may, for example, be
desirable to remove recycled points and/or IR depletion points);
- 'DoseSource.csv' gives the laboratory source dose rate and its variance;
- 'DoseEnv.csv' gives the dose rate to which the sample was exposed during burial.
We ran the function `AgeS_Computation()` with a prior age interval limited to between
10 ka and 100 ka for each sample (so that the bounds are far from the age values obtained using
arithmetic mean of equivalent doses, namely 37 ± 2 ka and 40 ± 2 ka, respectively). The dose-response





curves were fitted, as in *Analyst* in our previous study, with single saturating exponential functions
passing through the origin. All uncertainties, affecting both environmental and laboratory dose rates,
were included in the calculation, as is common practice in luminescence dating; however, the
covariance of ages was not modelled here, so the results are equivalent to those one would obtain by
running subsequent individual age calculations for each of the two samples.
To run the `AgeS_Computation()` function, the user must choose a model for the
distribution of individual equivalent doses around the central dose; the different options are Cauchy,
Gaussian or lognormal distribution (in the latter case, the central dose may be estimated either by the
mean or the median of the distribution). On top of saturation problems, Guérin *et al*. (2015b) also
identified dose rate variability as an important factor of dispersion in equivalent doses: the values of
the CDM overdispersion parameter for the $D_e$ distributions of the samples were equal to 29 ± 3 % and
35 ± 3 %, respectively. Consequently, to avoid problems of age underestimation, following the results
of laboratory-controlled experiments of Heydari and Guérin (2018), we did not use the Cauchy
distribution model. Instead, we modelled the equivalent dose distribution by a lognormal distribution
(one could also have chosen a Gaussian function) from which the mean (rather than the median) was
used to estimate the central dose. Indeed, Guérin *et al*. (2017) formally demonstrated that the median
of the lognormal distribution (as used in the CDM) is a biased estimator and leads to age
underestimates when dose rates are dispersed (see Heydari and Guérin, 2018, for experimental
confirmation of this demonstration).
After 5,000 iterations of 3 independent Markov Chains, we observed good convergence, as
seen in the Markdown document provided as supplementary material (for a discussion of the
convergence of the Markov Chains, the reader is referred to Philippe *et al*., 2019). The upper limit of
the 95% Credible Intervals (C.I.) for the Gelman and Rubin indexes of convergence (Gelman and Rubin,
1992) were all smaller than 1.05, also indicating satisfying convergence of the 3 independent Markov
Chains (here again, the reader is referred to Philippe *et al*., 2019, who suggested 1.05 as the maximum
acceptable value). The obtained 95% C.I. for the ages of samples FER 1 and FER 3 are [34.1; 43.3] ka
and [36.6; 47.8] ka, respectively (Fig. 1; Table 1) and are consistent with the ages obtained by Guérin
*et al.* (2015b) with a much simpler approach (unweighted arithmetic mean of equivalent doses). It
should be emphasised here that the two 95% C.I. for ages overlap. Fig. 2 shows a bivariate scatter plot
of a sample of observations from the joint posterior distribution of the two ages, as generated by the
Markov Chains; in such a plot, each point corresponds to one realisation of the ages of the two samples
investigated in the MCMC. Fig. 3 shows the corresponding probability densities for the ages estimated
jointly, based on kernel density estimates (KDE), and the marginal probability densities. No correlation
is observed on the joint probability density, which is symmetrical and bell-shaped. One can already
compare here the results obtained with this Bayesian model (lognormal-average) for sample FER 3
with the radiocarbon ages obtained independently for the same layer by Guérin *et al.* (2015b). The
95% C.I. for the 3 [14]C ages are bound by the interval [44.4; 47.3] ka, which means that the OSL and
radiocarbon ages are in good agreement, which was not the case when calculating the ages with the
CDM (38 ± 2 ka). Thus, even without further modelling, the 'BayLum' lognormal-average model seems
to provide OSL ages in better agreement with radiocarbon.
**4. Stratigraphic constraints**
Samples FER 1 and 3 belong to two different stratigraphic layers: sample FER 1 (Layer 7) lies
above sample FER 3 (Layer 5B), so we know that the age of sample FER 1 must be less than that of
sample FER 3. To encode this information, the function `AgeS_Computation()` takes as argument
the object `StratiConstraints`, which is a matrix whose size depends on the number of analysed
samples. First, the data in the DATA object (which is the output of the function





`Generate_DataFile()`) must be ordered in stratigraphic order from top to bottom: thus, in our
case the list of names used by the function `Generate_DataFile()` is FER 1, FER 3 (rather than FER
3, FER 1). Then, the stratigraphic matrix contains numbers equal to 0 or 1 indicating the applied bounds
to the age of each sample. The matrix contains a number of rows equal to the number of samples plus
one and a number of columns equal to the number of samples. The first row only contains 1 values,
which indicates that the lower age bound specified as prior information (10 ka in our example, *cf.*
section 3 above) when running the function `AgeS_Computation()` applies to all samples. Then,
for all *j* in {*2, …, Nb_Sample+1*} and all *i* in {*j, …, Nb_Sample*}, `StratiConstraints[j,i]`=1 if the
age of sample whose number ID is equal to *j-1* is smaller than the age of sample whose number ID is
equal to *i*. Otherwise, `StratiConstraints[j,i]`=0. In practice, in our case
`StratiConstraints [1,]` = (1,1), `StratiConstraints [2,]` = (0,1) (which means that
the age of sample FER 1 is not less than itself but is less than that of sample FER 3) and
`StratiConstraints [3,]` = (0,0) (which means that sample FER-3 is neither younger than
sample FER-1 nor itself). Note: in the markdown document provided as Supplementary Material, the
corresponding code lines are commented and perhaps make this description easier to follow.
Running the function `AgeS_Computation()` with this matrix of stratigraphic constraints
only marginally affects the ages, in this case, the 95% C.I. become [34.3; 42.9] ka and [38.1; 48.5] for
samples FER-1 and FER-3, respectively (Table 1). One can also look at the bivariate scatter plot of
observations from the joint posterior distribution (Fig. 4): one can see that this scatter plot is truncated
in the upper left-hand corner – illustrating the fact that the age of sample FER 1 can never be greater
than that of sample FER 3 (see Fig. 2 for comparison). By contrast, the KDE estimate (Fig. 5) also shows
a positive correlation but does not reveal the truncation (whereas the stratigraphic constraint imposes
a null probability for all pairs of ages above the 1:1 line).

**5. Dealing with multiple sources of errors through a covariance matrix**
**5.1. General considerations**
In the previous calculations, all the variance is treated as random, whereas common, systematic errors
should not allow solving stratigraphic inversions (they affect all ages in the same direction, although
to varying degrees). One of the main advantages of applying the models implemented in the 'BayLum'
package – contrary to other chronological modelling tools such as *OxCal* (Bronk Ramsey and Lee, 2013)
or *Chronomodel* (Lanos and Philippe, 2018) – lies in the possibility to include the structure of
uncertainties specific to OSL dating. For instance, a radiocarbon age is derived only from the ratio of
$^{14}$C to $^{12}$C (on top of which comes the more complex problem of calibration), whereas an OSL age
involves a large number of measurements, each with its uncertainty (Aitken, 1985; 1998). The OSL
measurements required for the determination of the palaeodose are relatively standardised through
the widespread use of the SAR protocol (Murray and Wintle, 2000; Wintle and Murray, 2006).
Conversely, there are several approaches – each with its equipment and standards – to determine the
various dose rate components. Given that these dose rates derive from different types of radiation
(alpha, beta, gamma and cosmic radiation) and are of various origins (mainly from potassium and the
uranium and thorium radioactive chains), there are many more contributions to the age uncertainty
from the dose rate term than from the palaeodose term, even though the size of the uncertainty on
dose rate is of the same order of magnitude as that on palaeodose – see for example Murray *et al.*,
2015). As a result, there are almost as many ways of estimating systematic and random uncertainties
as there are (combinations of) ways to determine dose rates. Combès and Philippe (2017) detailed the
mathematical formulation of the dose covariance matrix, which links the ages of several samples





measured using the same equipment and standards through common (systematic) errors (see also
Philippe *et al.*, 2019). Nevertheless, the equations provided in this article are somewhat difficult to
translate in practice; here, we propose to outline how we implement a covariance matrix adapted to
(one example of) the measurements leading to OSL age calculation at the IRAMAT-CRP2A laboratory
(Bordeaux). We emphasise that what follows is not prescriptive; it should be viewed as an example of
a model of uncertainties. For alternative ways of estimating systematic and random errors, for
example, due to different measurements of dose rates, the reader is referred to the discussion (section
268 7.1).

Here, we consider the case of a series of *n* sediment samples taken from one unique site and all
measured using the same equipment and standards. Let us consider the following relationship
between palaeodoses, dose rates and ages (Combès and Philippe, 2017):

$$(D_1, \dots, D_n) \sim \mathcal{N}\left((A_1\dot{d}_1, \dots, A_n\dot{d}_n), \Sigma\right)$$

where $D_i$ is a random variable modelling the unknown palaeodose of sample *i*, $\mathcal{N}$ is the symbol for a
Gaussian distribution, $A_i$ is the unknown age estimate of sample *i* (that we are trying to determine),
$\dot{d}_i$ the total dose rate to which this sample was exposed since burial ($\dot{d}_i$ is the observed dose rate, *i.e.*
the result of the measurements) and $\Sigma$ is the dose covariance matrix (for the full definition of the
model, we refer the reader to Combès and Philippe, 2017). This covariance matrix verifies, for all (i,j):

$$\Sigma_{i,j} = A_i A_j \theta_{i,j} \qquad \text{(Eq. 1)}$$

where $\theta$ is the matrix, the user needs to specify to run the calculations with 'BayLum'. It should be
noted here that by default when running age calculations with 'BayLum', the off-diagonal elements
are set to zero, *i.e.* the covariance in ages is not modelled.
Before entering the details specific to luminescence dating, let us consider a simple example of two
measurements $y_1 = \mu_1 + e_1 + f$ and $y_2 = \mu_2 + e_2 + f$ where $\mu_1$ and $\mu_2$ are fixed measurands and $e_1$, $e_2$ and
$f$ are all independent random errors from distributions with mean zero. The covariance of $y_1$ and $y_2$ is
the variance of $f$ (so the off-diagonal elements of the matrix are equal to this variance). For each
sample, the diagonal element of the corresponding covariance matrix is the sum of all the components
of variance for that sample. The variety of physical quantities to measure to determine dose rate, and
their relationship with the dose rate contributions, will now be discussed with this simple definition in
mind.

### 5.2. Implementation in practice

First, we detail the series of measurements carried out, and we introduce the corresponding
notations for the estimates and associated uncertainties. Table 2 summarises all physical units and
associated error standard deviations; as a general rule, we assume that all error terms are Gaussian
variables with the expected value (mean) equal to zero and a fixed, known standard deviation (see for
example Eq. 2 in Combès and Philippe, 2017). For clarity, in the following relative standard deviations
are described by the letter $\sigma$, while absolute standard deviations are denoted by *s*; moreover, each
standard deviation corresponding to random errors (*i.e.*, when the error varies from sample to sample)
is identified by the letter *i* in the subscript. The absence of this letter in the subscript indicates that the
measurement error affects all samples.

### 5.2.1. Equivalent doses and OSL measurements

Equivalent doses are determined from OSL measurements performed on a luminescence
reader equipped with a radioactive beta source, whose dose rate and associated relative standard



deviation of the error, noted $\dot{d}_{\text{lab}}$ and $\sigma_{\text{lab}}$, are known. There are several ways the latter term can be
determined; in its simplest form, it includes the standard deviation of the error on the absolute dose
absorbed by the standard reference material (in our case calibration quartz provided by DTU Nutech,
*cf.* Hansen *et al.,* 2015) and an error term due to replicate measurements of several aliquots of this
calibration material. Using a large number of measurements repeated in time, as suggested by Hansen
*et al.* (2015), may somewhat complicate the matter, but this goes beyond the scope of the present
study.

310       In practice, regeneration doses are delivered by irradiating the aliquots for a given duration (in
s). This duration is converted to absorbed energy dose (Gy) by multiplication with the source dose rate
(Gy.s$^{-1}$). Strictly speaking, the error on the source dose rate affects all regeneration doses, and so this
error term should appear in the dose/luminescence relationship (right side of the directed acyclic
graph shown in Fig. 7 of Combès and Philippe, 2017). However, it is common practice in the field of
luminescence dating to first calculate an equivalent dose in seconds of irradiation for each aliquot,
then convert this to Gy and calculate an average (or determine another central parameter such as with
the CDM), and only then consider $\sigma_{\text{lab}}$. This is what led, *e.g.*, Jacobs et al. (2008), to exclude the
associated standard deviation from the total OSL age uncertainties, to test the assumption of a time
gap between two series of ages. Here, for simplicity, we take the same route, and hence the relative
error on the laboratory source dose rate becomes a relative, systematic error on the equivalent doses.

321       One may thus write that the error on the dose $D_i$ arising from the calibration of the source
follows a Gaussian distribution with mean 0 and variance $(\sigma_{\text{lab}} A_i \dot{d}_i)^2$.

### 5.2.2. Dose rates

325       When it comes to the dose rate term, here we restrict ourselves to the case of coarse quartz
grains measured after HF etching to remove the alpha dose rate component: the total natural dose
rate is the sum of an internal dose rate, external beta and gamma dose rates, and cosmic dose rates.

**Cosmic dose rates**

329       We consider that cosmic dose rates are determined following Prescott and Hutton (1994)
based on the burial depth of the dated samples, which may be different from the present-day thickness
of the overburden. As a result, the error on cosmic dose rate estimates depends on the error
estimation of this effective burial depth since the dated sediment was deposited. Because the
relationship between cosmic dose rates and burial depths is not linear, and because the error on this
burial depth may not be systematic (*e.g.* in cases where successive, yet of unknown duration, erosion
and deposition events happened between the deposition of superimposed sedimentary layers – see
Aitken, 1998, p. 65, for a discussion) even at the scale of a site the error associated to cosmic dose
rates cannot easily be treated as systematic. For *i={1,…,n}*, $\dot{d}_{\text{cosmic},i}$ and $s_{\text{cosmic},i}$ denote the estimate
of the average cosmic dose rate to which sample *i* has been exposed and its associated standard
deviation.

**Beta dose rates**

341       We consider the beta dose rates as determined from concentrations (or activities) of $^{40}$K and
in radioelements from the U- and Th- decay chains, converted to dose rates using specific conversion
factors (*e.g.,* Guérin *et al*., 2011). At the IRAMAT-CRP2A laboratory, these concentrations are usually
determined with low-background, high-resolution gamma-ray spectrometry following Guibert and
Schvoerer (1991). The simplest case is that of $^{40}$K, since only one peak is used (at 1.461 MeV); the





concentration in sample *i*, denoted [K]$_i$ is equal to the concentration in the standard multiplied by the
ratio in count rates (the count rate observed for the investigated sample is divided by the count rate
observed for a reference material). Thus, we consider in this paper that the standard deviation of the
error on the $^{40}$K concentration includes three components: the standard deviation of the error on the
concentration in the standard, and the counting uncertainties both on the standard and on the
measured sample. The counting uncertainties are calculated, assuming Poisson statistics. Of these
three sources of errors, only one is treated as random – namely the counting uncertainty of the sample;
the other two standard deviations (corresponding to the counting of the standard and to the error of
the radioelement concentration in the standard) are quadratically summed and considered as a
systematic source of error. One considers for sample *i* the beta dose rate from potassium $\dot{d}_{\beta,K,i}$ – after
correction for grain size-dependent attenuation using the factors from Guérin *et al.*, (2012a); and for
moisture content following Nathan and Mauz (2008) (see the discussion section below regarding
uncertainties on these correction factors). Neglecting uncertainties in the dose rate conversion factors,
we call $\sigma_{K,i}$ the relative random standard deviation of the error on the $^{40}$K concentration; its systematic
counterpart $\sigma_K$ is common to all samples. It should be emphasised here that systematic errors on
radioelement concentrations, although being shared by all samples, will affect all ages in the same
direction but not necessarily by the same amount (even in relative terms, contrary to the error on
laboratory beta source calibration) because the relative contribution of beta dose rate from potassium
to the total dose rate may vary from one sample to another. The beta dose rates from the U- and Th-
series come from a number of radioelements in the corresponding chains; here, for simplicity we
consider each series to be in secular equilibrium (this is generally the case for $^{232}$Th but may not be
true for the U-series, see, *e.g.* Guibert *et al.*, 1994; 2009; Lahaye *et al*, 2012). Thus, for each sample,
the concentrations in $^{238}$U and $^{232}$Th are converted to dose rate contributions denoted $\dot{d}_{\beta,U,i}$ and
$\dot{d}_{\beta,Th,i}$. In contrast to the case of $^{40}$K, the analysis of the high-resolution spectra for these radioactive
chains is based on a number of primary gamma rays; more specifically, a weighted mean of the
concentrations determined from each ray included in the analysis (after taking interference into
account) is calculated to estimate the concentration of U (resp. Th). As a result, the standard deviations
of the errors on these concentrations are the contributions of two sources: the relative standard
deviation on the concentrations of the standards correspond, on the one hand, to systematic sources
of errors and are denoted $\sigma_U$ and $\sigma_{Th}$; conversely, all other relative standard deviations (arising from
the counting of the standards and of the sample) are treated as random and denoted $\sigma_{U,i}$ and $\sigma_{Th,i}$.
**Internal dose rates**
Unless the internal radioelement concentration is experimentally determined (in which case
one needs to consider both systematic and random sources of error for each sample, as is done for
beta dose rates), some have suggested using a fixed internal dose rate of 0.06 ± 0.03 Gy.ka$^{-1}$ (Mejdahl,
personal communication to Murray, based on Mejdahl, 1987). In this case, we may assume that the
dated quartz grains are all of the same origin, and have the same internal radioelement concentration;
as a result, we associate a systematic standard deviation $s_{int}$ with the internal dose rate $\dot{d}_{int}$.
**Gamma dose rates**
Gamma dose rates $\dot{d}_{\gamma,i}$ may be determined, as beta dose rates, from K, U and Th
concentrations in the sediment. In this case, the reader is referred to the corresponding section above.
However, it is relatively frequent, in the case of heterogeneous configurations at the 10 cm scale, that
gamma dose rates received by the samples do not correspond to the infinite matrix gamma dose rates
of the samples (see for example large gamma dose rate variations at the interface between sediment
and bedrock in a cave reported by Guérin *et al.*, 2012b: Fig. 7). In such contexts, gamma dose rates





may be determined by *in situ* measurements with $Al_2O_3$:C artificial dosimeters: these dosimeters are
measured with green-light stimulation and their calibration is based on a block of homogeneous bricks
located in the basement of IRAMAT-CRP2A (Richter *et al.*, 2010; Kreutzer *et al.*, 2018). Two sources of
relative errors are taken into account: a random standard deviation ($\sigma_{\gamma,i}$) accounting for measurement
uncertainties, and a shared calibration error including both standard deviations on (i) the true gamma
dose rate in the block of bricks and on (ii) the measurement of the dosimeters irradiated inside the
block for calibration of the source ($\sigma_\gamma$).
**Water content**
To account for the effect of water on dose rates, one commonly considers the following
equation (Zimmerman, 1971; Aitken, 1985):
$$\dot{d}_{\beta,i} = \frac{\dot{d}_{\beta,i,\mathrm{dry}}}{1 + x_\beta WF_i},$$
where $\dot{d}_{\beta,i,\mathrm{dry}}$ is the beta dose rate in the dry sediment, $WF_i$ represents the effective mass fraction of
water in the sediment during burial, and $x_\beta$ is a water correction coefficient accounting for the fact
that water absorbs more beta dose than typical sedimentary elements, due to lower atomic numbers
(Nathan and Mauz, 2008). A similar equation applies to gamma dose rates, with a corresponding factor
$x_\gamma$ (see Guérin and Mercier, 2012). The determination of the water content in the sediment over time
is a very difficult task as it involves many different parameters, including past rainfall. One commonly
employed solution is to measure the water content at the time of sampling and assume it to be
representative of that in the past; measuring the water content at saturation may then be a solution
to evaluate an upper limit to this value; and depending on the context one may also propose a lower
limit to the water content. One then obtains a way of quantifying the standard deviation of the error
on the water content, although necessarily imperfect (see Nelson and Rittenour, 2015, for a
discussion). Neglecting uncertainties on the water correction factors ($x_\beta$ and $x_\gamma$) and calling $s_{WF,i}$ the
absolute standard deviation of the mass fraction $WF_i$ for sample *i*, one may write:
$$s_{\beta,H_2O,i} = \dot{d}_{\beta,i} \frac{s_{WF,i}}{1 + x_\beta WF_i}$$
where $s_{\beta,H_2O,i}$ is the standard deviation of the beta dose rate for sample *i* due to the uncertainty on
its water mass fraction.
Similarly, one may write:
$$s_{\gamma,H_2O,i} = \dot{d}_{\gamma,i} \frac{s_{WF,i}}{1 + x_\gamma WF_i}$$
where $s_{\gamma,H_2O,i}$ is the standard deviation of the gamma dose rate for sample *i* due to the uncertainty on
its water mass fraction. As a result,
$$s_{\gamma,H_2O,i} = \frac{\dot{d}_{\gamma,i}}{\dot{d}_{\beta,i}} \frac{1 + x_\beta WF_i}{1 + x_\gamma WF_i} s_{\beta,H_2O,i}.$$
To simplify the following equations, which are meant to be those used in practice, we introduce the
relative standard deviation of the beta dose rate due to water content errors ($\sigma_{\beta,H_2O,i}$) and a parameter
called $\lambda_i$ defined by:
$$\lambda_i = \frac{1 + x_\beta WF_i}{1 + x_\gamma WF_i}$$




**The $\theta$ matrix**

With these considerations in mind on errors and their nature, the corresponding $\theta$ matrix (Eq. 1) to model these uncertainties is a square matrix containing one line (and column) per sample. The diagonal elements correspond to the sum of a term arising from the error on the laboratory source dose rate ($\dot{d}_i^2 \sigma_{\text{lab}}^2$) and the total dose rate variance for each sample, for each $i$:

$$\theta_{i,i} = \dot{d}_i^2 \sigma_{\text{lab}}^2 + s_{\text{cosmic},i}^2 + \dot{d}_{\beta,\text{U},i}^2 (\sigma_{\text{U},i}^2 + \sigma_{\text{U}}^2) + \dot{d}_{\beta,\text{K},i}^2 (\sigma_{\text{K},i}^2 + \sigma_{\text{K}}^2) + \dot{d}_{\beta,\text{Th},i}^2 (\sigma_{\text{Th},i}^2 + \sigma_{\text{Th}}^2) + s_{\text{int}}^2$$
$$+ \dot{d}_{\gamma,i}^2 (\sigma_{\gamma,i}^2 + \sigma_\gamma^2) + (\dot{d}_{\beta,\text{U},i} + \dot{d}_{\beta,\text{K},i} + \dot{d}_{\beta,\text{Th},i} + \lambda_i \dot{d}_{\gamma,i})^2 \sigma_{\beta,\text{H}_2\text{O},i}^2.$$

This long list of variance terms may seem rather complicated, but it corresponds to the total variance arising from the laboratory beta source calibration, the errors on cosmic dose rates, environmental beta dose rates internal dose rates, gamma dose rates, and finally the error arising from uncertainties in water content. In other words, we can also write

$$\theta_{i,i} = \dot{d}_i^2 \sigma_{\text{lab}}^2 + s_{\dot{d}_i}^2 \qquad \text{(Eq. 2)},$$

where $s_{\dot{d}_i}^2$ is the variance of the dose rate to which sample $i$ was exposed to during burial (it is the square of the uncertainty appearing next to the dose rate value in every luminescence dating article; in our example, this term is the second one in the files DoseEnv.csv provided in Supplementary Material).

Then, for $i \neq j$:

$$\theta_{i,j} = \dot{d}_{\gamma,i} \, \dot{d}_{\gamma,j} \sigma_\gamma^2 + \dot{d}_{\beta,\text{U},i} \dot{d}_{\beta,\text{U},j} \sigma_\text{U}^2 + \dot{d}_{\beta,\text{K},i} \dot{d}_{\beta,\text{K},j} \sigma_\text{K}^2 + \dot{d}_{\beta,\text{Th},i} \dot{d}_{\beta,\text{Th},j} \sigma_\text{Th}^2 + s_{\text{int}}^2 + \dot{d}_i \, \dot{d}_j \sigma_{\text{lab}}^2 \qquad \text{(Eq. 3)},$$

which characterises the amount of correlation between the doses of samples $i$ and $j$, multiplied by their ages. The $\theta$ matrix, like the dose covariance matrix $\Sigma$, is a symmetric matrix. The diagonal members correspond to individual variances, while the non-diagonal terms express the fact that systematic, shared errors link the measurements of the series of samples. As a result, running the functions `AgeS_Computation()` and `Age_OSLC14()` with a $\theta$ matrix in which all non-diagonal members are set to zero would be equivalent to running the same functions without the correlation matrix, or running the function `Age_Computation()` independently for each sample – in which case all sources of error are treated as random.

**5.3. Examples**

**5.3.1. An illustrative, simplistic example without stratigraphic constraints**

For illustration purposes, first, we did not apply stratigraphic constraints. We started with a simplistic $\theta$ matrix containing in the diagonal the real error variances (Eq. 2) as determined by Guérin *et al.* (2015); the $\sigma_{\text{lab}}$ value was equal to 0.02 (2% relative standard deviation of the calibration of the laboratory beta source). The simplification comes from the off-diagonal members, for which in Eq. (3) we set all $s$ and $\sigma$ values equal to 0, except for the $\sigma_{\text{lab}}$ value, set to 0.05. Obviously, this is not self-consistent, but it corresponds to (i) random and systematic errors of approximately the same magnitude (in practice, these two sources of errors are of the same order of magnitude – a few %) and (ii) the simplest form of systematic errors. Indeed, in such a case, all ages are affected by the same relative amount in the same direction.

Here again, after 5,000 iterations of 3 independent Markov Chains, we observed good convergence. The obtained 95% C.I. are [33.9; 43.8] and [36.7; 48.1] ka for samples FER 1 and FER 3, respectively. Fig. 6 shows bivariate scatter plots corresponding to the sampling of the Markov Chains



for the ages of samples FER 1 and FER 3 (which are calculated simultaneously) and Fig. 7 displays the
KDE together with the marginal probability densities. This set of figures illustrate the reason for the
generation of the two type of figures: the bivariate scatter plot is most appropriate to visualise the
effect of stratigraphic constraints (Fig. 4 above), whereas probability density figures best illustrate the
effect of modelling systematic errors. Indeed, as can be seen, there is a positive correlation between
the ages of samples FER 1 and FER 3: the greater the age of sample FER 1, the greater is the mean age
of sample FER 3. In other words, if the age of sample FER 1 was underestimated, then in all likelihood,
so would be the age of sample FER 3. Furthermore, the length of the C.I. for the age of each sample is
slightly larger than without modelling the covariance (*cf*. Table 1), *i.e.* modelling the covariances
slightly increases the age uncertainties. However, the positive correlation of ages has other, direct
consequences.
First, let us suppose that we have no knowledge of a stratigraphic link between the two
investigated samples, and wish to test the hypothesis that sample FER 1 is younger than sample FER 3.
The credibility of such an assumption can be tested using the function `MarginalProbability()`
of the 'Archaeophases' R package (Philippe and Vibet, 2020) devoted to the analysis of MCMC chains
for chronological inference. Without using the covariance matrix, the credibility of this hypothesis is
0.83; with the simplistic $\theta$ matrix, the credibility becomes 0.94; in other words, modelling the age
covariance reflects more faithfully the measurements and their uncertainties for such tests.
The second consequence concerns the duration of a hypothetical phase that would encompass
the deposition of sample FER 1 and that of sample FER 3. Indeed, since the ages vary together in the
MCMC, the duration of such a phase should be smaller when modelling the covariance than when all
the variance in ages is treated as random. Indeed, we could verify this assertion using the function
`PhaseStatistics()` of 'ArchaeoPhases' (Philippe and Vibet, 2020): with the simplistic covariance
matrix, the 95 % C.I. for the duration of this phase is [-1.4; 9.7] ka, whereas it is [-0.6; 7.6] ka when the
ages are calculated using the simplistic $\theta$ matrix.

### 5.3.2. A real example, including stratigraphic constraints

In a real case, since the relative contributions of the different dose rate components vary from
one sample to another, the correlation will be less pronounced. For more realistic calculations of the
ages of samples FER 1 and FER 3, we took the same values as above for the diagonal terms of the $\theta$
matrix (Eq. 2); on the other hand, for the non-diagonal, covariance terms, we used the following values:
$\sigma_{\mathrm{lab}} = 0.02$ (which corresponds to the experimentally determined calibration standard deviation,
including the uncertainty of the dose delivered to calibration quartz; Hansen *et al*., 2015), $\sigma_{\mathrm{K}} = 0.012$,
$\sigma_{\mathrm{U}} = 0.007$ , $\sigma_{\mathrm{Th}} = 0.007$ (for these values, which also include counting of the standards used, the
reader is referred to Guibert *et al.*, 2009; Guibert, 2002), and $s_{\mathrm{int}} = 0.003$ Gy.ka⁻¹. We provide as
Supplementary Information a calculation spreadsheet allowing to build the covariance matrix,
intended for adaptation to the user-specific needs.
At the site of La Ferrassie, the uncertainties associated with the gamma dose rate observations
are more complex. $Al_2O_3$:C dosimeters were placed at the end of 25 cm long aluminium tubes and
inserted horizontally in the stratigraphic section at the location of sediment sampling. In an ideal case,
sediment should be uniform in a horizontal plane; however, for samples FER 1 and FER 3 only a rather
thin layer of sediment remained against the cliff wall (the layers of the sample were not present at the
site in any other location), which resulted in the dosimeters being inserted either in the karstic cliff
(the limestone contains little radioelements compared to the sediments, as shown in Fig. 5 of Guérin
*et al*., 2015b) or at the interface between the cliff and the sediment. As a result, we took for $\dot{d}_{\gamma,i}$ the
average between the gamma dose rates measured *in situ* (which underestimate the real gamma dose





rate because the effect of the cliff is over-represented) and the gamma dose rates derived from the K,
U and Th concentrations in the samples. The associated standard deviation, $\sigma_{\gamma,i}$, was calculated as the
difference between these two extreme values divided by 4, so that the 95% C.I. covers all possible
values. As this standard deviation is much larger than the analytical uncertainties, we neglected the
latter and considered $\sigma_{\gamma,i}$ to characterise random sources of errors since each sample has a different
environment and may be more or less far from the cliff.
The samples FER 1 and FER 3 are directly above and below, respectively, the Châtelperronian
layer at the site (layer 6). Sample FER 2 from this layer being poorly bleached, it is at present impossible
to model with 'BayLum'. However, an alternative to estimate the age of FER 2 consists of supposing
that it has a uniform prior probability density between the ages of samples FER 1 and FER 3:
$$P(A_2|data) \sim \iint \frac{\mathbb{I}_{[A_1;A_3]}}{A_3 - A_1} \pi(A_1, A_3|data) dA_1 dA_3$$

where $A_i$ is the age of sample $i$, $\mathbb{I}_{[A_1;A_3]}$ is the indicator function between $A_1$ and $A_3$, and
$\pi(A_1, A_3|data)$ is the posterior joint density of $A_1$ and $A_3$ knowing the data (*i.e.* the density estimated
with 'BayLum'). Doing so (see the markdown file for the corresponding code lines), working from the
output of 'BayLum' one obtains a 95% C.I. of [36; 46] ka, which can be compared with the confidence
interval of [36; 48] ka obtained by Guérin et al. (2015) with minimum age modelling.
**6. Integration of independent chronological data (radiocarbon)**
The 'BayLum' package also offers the possibility to include radiocarbon ages in the chronological
models (Philippe *et al.*, 2018); more specifically, radiocarbon ages are calibrated within 'BayLum', using
the function `AgeC14_Computation()` or `Age_OSLC14()` (in the latter case the function
necessitates at least one OSL age calculation). Introducing covariance matrices to account for
systematic errors on OSL data does not reduce the OSL age uncertainties; however, it becomes
particularly useful to correct for estimation biases when more precise ages, unaffected by these
systematic errors, are integrated into the models. To illustrate this, we decided to construct two
models constraining the age of FER 3; for illustration purposes, in this section, we used the simplistic
$\theta$ matrix described above in section 5.3.1. In the first case, we constrained the age of this sample by
imposing that a 'young' radiocarbon age (young compared to the age of sample FER 3 considered
alone) has an age greater than sample FER 3. In practice, we arbitrarily took a radiocarbon age of
38,000 ± 400 BP, which corresponds to [37.6; 39.9] ka cal. BP (95% C.I. using the IntCal20 curve, Reimer
*et al.*, 2020; the calibration was performed using 'BayLum', see Philippe et al., 2018). Naturally, the
credible intervals (both 68% and 95%) for sample FER 3 are shifted towards younger age values (*cf.*
truncation of the scatter plot illustrated in Fig. 3). So do the credible intervals for sample FER 1, since
the ages of the two OSL samples are close to each other even when considered independently of
radiocarbon data (in other words, the radiocarbon age 'pushes' the age of sample FER 3, which in turn
'pushes' the age of sample FER 1). In practice, the 95% C.I. become [33.3; 41.2] ka and [36.9; 42.3] ka
for samples FER 1 and FER 3, respectively. It can be noted here that in such a case the precision of the
age of sample FER 3 is increased (*i.e.* the length of the C.I. is much smaller than without the constraining
radiocarbon age). More interestingly, in the second case, we constrained the age of sample FER 3 by
imposing that an 'old' radiocarbon age (old compared to the age of sample FER 3 considered alone)
has an age younger than sample FER 3. In practice we – again, arbitrarily – took a radiocarbon age
equal to 44,000 ± 400 BP, which corresponds to [45.4; 47.4] ka cal. BP (95% C.I.). Here again, the effect
on the age of sample FER 3 is straightforward: the credible intervals are shifted towards older ages
(the 95% C.I. for the age of sample FER 3 becomes [45.7; 51.2] ka). Perhaps less intuitive is the effect
on the age of sample FER 1, which is not directly constrained by radiocarbon: because the ages of the



three samples are estimated jointly, and because of the systematic errors on the OSL ages, the age of
sample FER 1 is also shifted towards older ages: the corresponding 95% C.I. becomes [36.7; 45.8] ka.
**7. Discussion**
**7.1. Differing ways of estimating dose rates**
Every laboratory uses its specific equipment and calibration standards; if similar equipment as
described above is used, then only the values of the different terms need be changed. This case is
particularly relevant for equivalent dose measurements, and hence the term $\sigma_{lab}$ associated with $\dot{d}_{lab}$.
Conversely, for dose rate determination, several other experimental devices and techniques are
commonly used. If beta and/or gamma dose rates are determined based on the determination of
concentration in K, U and Th, (for example by mass spectrometry, neutron activation, etc.), then the
situation is similar as that described for beta dose rates above.
Counting techniques (alpha, beta, and gamma in the case of the threshold technique: Løvborg
*et al*., 1974) may also be used for beta and gamma dose rate estimation. In the case of beta counting,
the conversion factor from count rate to dose rate depends on the emitting radioelement
(Ankjærgaard and Murray, 2007; see also Cunningham *et al*., 2018). This dependency is a source of
error that may not be characterised by a systematic error (so there is no contribution to the dose
covariance matrix). The data acquired with field gamma spectrometers may be analysed in two ways:
the 'window' technique (see, *e.g.*, Aitken, 1985) corresponds to classical spectrometry analysis; in this
case, the structure of uncertainties is the same as that for beta dose rates determined from high-
resolution gamma spectrometry (Eq. 3). On the other hand, threshold techniques consist of taking
advantage of proportionality between gamma dose rates and (i) the number of counts recorded per
unit time above a threshold (Løvborg and Kirkegaard, 1974) or (ii) the energy deposited per unit time
above a threshold (energy threshold: Guérin and Mercier, 2011; Miallier *et al*., 2009). In the former
case, the conversion from count rate to dose rate depends on the emitting radioelement, so no
systematic error term may be isolated. Conversely, in the latter case (energy threshold), this
dependency is negligible (Guérin and Mercier, 2011). As a result, the error on the dose rate of the
calibration standard may be considered as systematic, and thus contribute one term in the non-
diagonal elements of the covariance matrix.
**7.2. Error terms neglected in this study**
As mentioned earlier in the section devoted to dose rate uncertainties, there are many
possibilities to quantify, but also to consider errors on dose rate measurements; one could mention
here the uncertainties on attenuation factors and water correction factors. However, both of these
factors are dependent on the infinite matrix assumption: attenuation in grains implies that something
other than the grains does not attenuate radiation; water correction factors are often calculated
assuming a homogeneous mixture of water and other sedimentary components (Zimmerman, 1971;
Aitken and Xie, 1990; note: the composition of the sediment also necessarily affects the ratios of
electron stopping powers and photon interaction cross-sections – see Nathan and Mauz, 2008, for a
discussion). Limitations of this infinite matrix assumption, which is not met in sand samples at the scale
of beta dose rates, have already been pointed out (Guérin and Mercier, 2012; Guérin *et al.*, 2012a;
Martin *et al.,* 2015). Consequently, it seems that routine determination of a realistic standard deviation
of the attenuation and water content correction parameters is not straightforward.
Dose rate conversion factors were assumed above to be known without error; however,
estimation errors do affect half-lives, emission probabilities, average emitted energies, etc. Liritzis *et*
*al.* (2013) took these uncertainties into account to estimate standard deviations of the dose rate




conversion factors (in practice, these standard deviations amount to ~1% for K dose rates, ~2% for U
and ~2% for Th). These standard deviations could be included as sources of systematic errors when the
contributions of K, U and Th are determined separately (note: when this is not the case, as when
dosimeters are used for gamma dose rate estimation, or when beta counting is implemented for beta
dose rate assessment, these sources of errors should be treated as random).
In this study, we worked with coarse grain quartz extracts that had been etched with HF to
remove the alpha-irradiated part of the grains. This being said, if alpha dose rates are taken into
account, then the situation becomes similar to that of beta dose rates treated above; however, the
sensitivity to alpha irradiation must then be taken into account. It is rather frequent in such a case to
use published values from the literature (e.g., Tribolo *et al*., 2001; Mauz *et al*., 2006). Depending on
the geological origin of the quartz (one or more sources), one may then assume either systematic or
random errors on the alpha sensitivity.

### 7.3. Publication habits and re-analysis of previously published ages

Compared to other statistical models for OSL dating, the Bayesian models implemented in
'BayLum' appear rather complicated, at least partly because modelling starts from the measured OSL
data. By comparison, the input data to the CDM or the Average Dose Model (ADM: Guérin *et al*., 2017)
are lists of equivalent doses and associated uncertainties, which means that OSL measurements have
already been analysed to derive equivalent doses. Combès *et al*. (2015) argued that their complete
model (implemented in 'BayLum'), relating all the variables to one another, produces a more
homogeneous and consistent inference compared to consecutive inferences (and indeed, when
approaching saturation, *i.e.* when equivalent doses and associated uncertainties can hardly be
parameterised, Heydari and Guérin (2018) demonstrated the advantage of 'BayLum' models compared
to parametric models such as the CDM and ADM in particular settings. However, working with lists of
equivalents doses and uncertainties – or even with estimates of central doses and associated
uncertainties – taken as observations would make the Bayesian modelling proposed in 'BayLum' and
described in this paper more straightforward and transparent. Such an approach, called the 'two-steps'
model by Combès and Philippe (2017; see also Millard 2006a, 2006b, for earlier, similar models), would
also offer the advantage of allowing re-analysis of already published data to derive more precise
chronologies. However, for this purpose the breakdown of all uncertainties and related standard
deviations of errors is needed; nowadays, providing such key information for the modelling is not in
the publication habits of the luminescence dating community. That being said, with the growing
number of meta-analyses of previously published data, and the availability to use models such as
BayLum to combine measurements with systematic errors, these habits might evolve in the future.

### 7.4. Notes of caution

As always when working with statistical models, one should first and foremost evaluate the
measured data in the light of sampling context. We already mentioned the importance of grain
selection (section 2.2.); but, perhaps more importantly, and especially since users of 'BayLum' have to
make modelling choices (*e.g.*, regarding the dose-response curves fitted to OSL measurements or the
distribution of individual equivalent doses around the central dose), it is crucial to carefully examine
data and assess their quality before building potentially sophisticated models.
We would like here to emphasise a few warnings regarding modelling samples in stratigraphic
constraints, and the association of ages obtained by different methods. We would advise users, before
combining, *e.g.* radiocarbon and OSL ages, first to thoroughly examine the corresponding datasets
independently: how were the data produced (with which experimental procedure)? Are the provided
uncertainties reliable (or is there an unrecognised source of error that should be included in the




evaluation of uncertainty)? Users are also encouraged to examine the consistency of results produced
by each method, in light of the stratigraphy. In a second stage, before modelling of independent ages,
we would recommend assessing the consistency of these datasets – do they (at least broadly) agree?
And if not, can a parsimonious explanation be found? For example, it is rather common, when
performing Bayesian modelling with tools such as *OxCal*, to observe a large fraction of ages considered
as outliers; such observations should urge users to examine their data again and come up with likely
explanations (note: to this date, no outlier model has been developed for the OSL ages in 'BayLum').
When it comes to imposing ordering constraints between ages as a result of stratigraphic observations,
it is, of course, essential to leave no doubt about the validity of these stratigraphic constraints (the
results of a model depend on the assumptions that are made, and the order in ages is a very strong
constraint). Perhaps more importantly, even when stratigraphic constraints are valid, it is possible that
applying them will not improve the statistical inference.
A simple example to illustrate this point is that of two superimposed, distinct layers (so that a
stratigraphic order is clear) whose true ages are equal (or in practice, for which the age difference is
negligible compared to the typical uncertainties of the implemented dating method). In such a case,
modelling the ages with stratigraphic constraints is likely to result in a loss of accuracy (the age of the
older layer will be overestimated, and that of the younger layer underestimated) compared to a model
where no stratigraphic constraints are imposed. Future developments of the 'BayLum' package might
include the possibility to test different modelling scenarios by comparing the agreement between the
observations and the posterior probability densities, for example using the Bayes Information Criterion
(BIC).
**8. Conclusion**
New models for building chronologies based on OSL, with the possibility to incorporate radiocarbon,
have been proposed in the literature (Combès *et al.*, 2015; Combès and Philippe, 2017). These models
have been demonstrated to improve the chronological inference based on OSL data and in particular,
the accuracy of OSL ages (Guérin *et al.*, 2015; Heydari and Guérin, 2018). The R package 'BayLum' was
developed to implement these models; Lahaye *et al.* (2018), Carter *et al.* (2019), Heydari *et al.* (2020)
and Heydari et al. (in review) have used some of them to establish the chronologies of sedimentary
sequences dated by OSL, resulting in generally more precise chronologies.
In this article, we have presented a case study on how to build simple models and observe output data,
in particular through bivariate plots of age probability densities. Then, we have shown how to include
stratigraphic constraints in the models; we have described how to fill the covariance matrices to
account for systematic errors in OSL age estimation; and we have shown the effect of including
independent age information in the models, namely radiocarbon ages. Different tools to visualise and
further analyse the output of 'BayLum' were demonstrated.
As a result, it is now possible to make use of various information often available in practice when dating
stratigraphic sequences. Age inferences based on OSL and independent data (*e.g.,* radiocarbon) in
stratigraphic constraints are expected to gain in accuracy, precision and robustness, through the
application of such Bayesian models.

**Acknowledgements**
The authors thank Andrew Millard, Rex Galbraith and an anonymous referee for helpful comments on
a previous version of this article. This study received financial supports of the Région Aquitaine (in



particular through the CHROQUI programme) and of the LaScArBx project (project n◦ ANR-10-
LABX-52). M. Autzen, and J.P. Buylaert received funding from the European Research Council (ERC)
under the European Union's Horizon 2020 research and innovation programme ERC-2014-StG 639904-
RELOS.

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



**Figures**

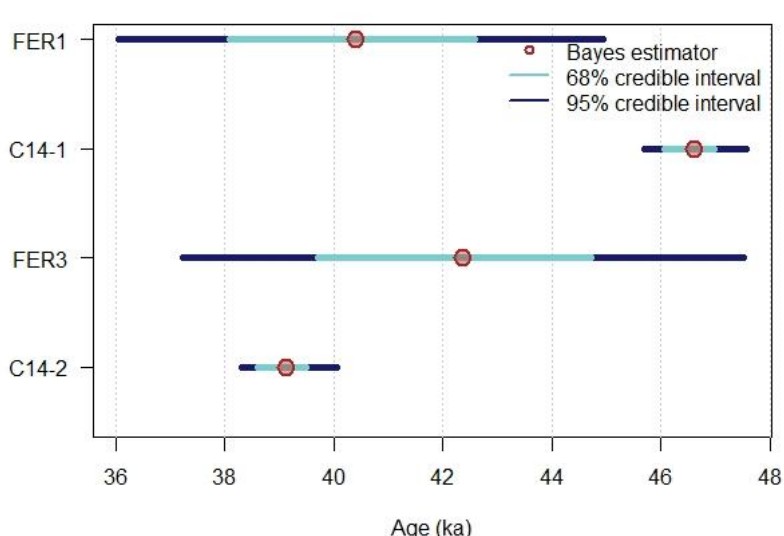


**Fig. 1:** Age estimates for OSL samples FER 1 and FER 3. The red circles indicate the Bayes estimates of
the age (*i.e.* the most likely values) for each sample; the cyan and blue bars represent the 68% and
95% credible intervals, respectively. For the two radiocarbon ages (C14-1 and C14-2), the reader is
refereed to section 6.




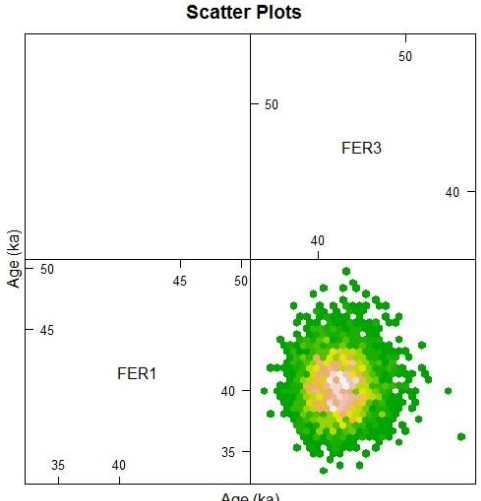


**Fig. 2:** Bivariate scatter plot as hexagon plot presentation of a sample of observations from the joint
posterior distribution of the two OSL ages considered independently (no stratigraphic constraints, no
off-diagonal members in the covariance matrix). In such a plot, each point corresponds to one
realisation of the ages of the two samples generated by the MCMC. Note: the reason for having this
figure in the cell of an array is not visible here; it becomes useful when calculating ages for more than
2 samples, in which case for each pair of samples, a similar plot appears in the appropriate cell.




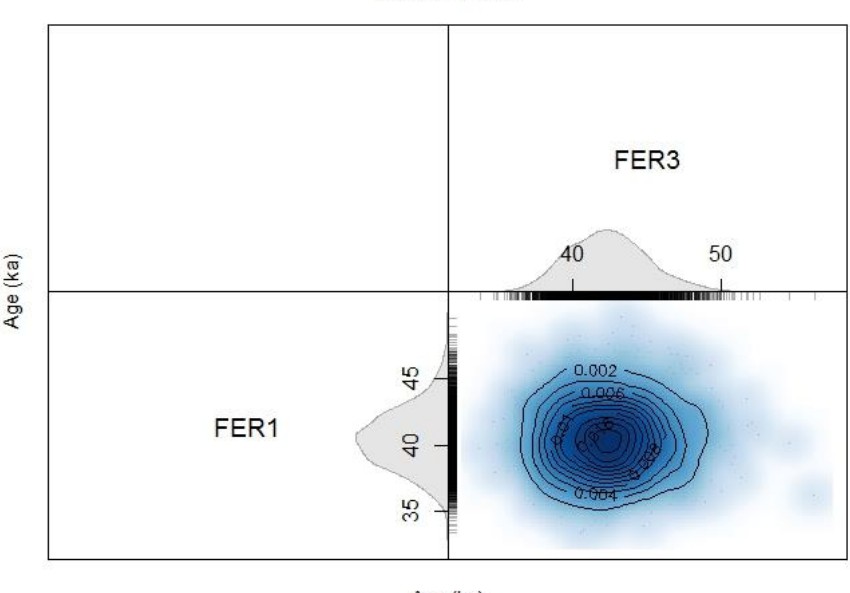


**Fig. 3:** Probability densities for the OSL ages estimated jointly with the same model as that used to
generate Fig. 2, based on Kernel Density Estimates (KDE), and marginal probability densities. The bell-
shape and symmetry of the scatter plot indicate the absence of correlation between the two ages.


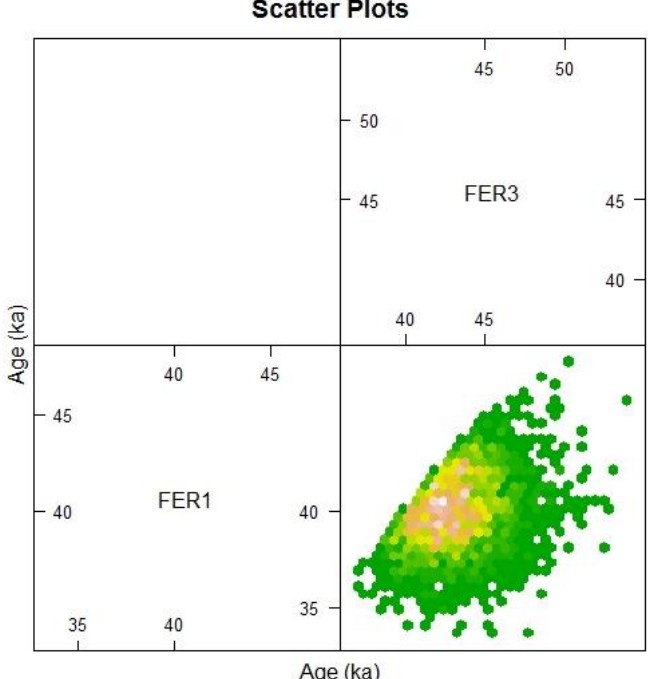

**Fig. 4:** Bivariate scatter plot from the joint posterior distribution of the ages of samples FER 1 and FER
3 when a stratigraphic constraint is applied (sample FER 1 is younger than sample FER 3) but with no
off-diagonal members in the covariance matrix. The truncation in the upper-left hand corner scatter
plot indicates the effect of the stratigraphic constraint.





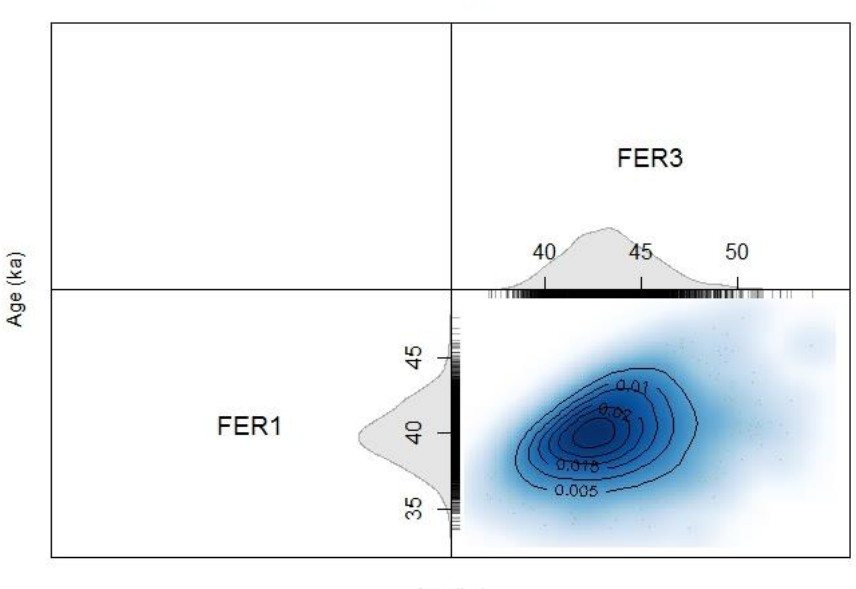

**Fig. 5:** Probability densities for the OSL ages estimated jointly, using the same model as that
implemented to generate Fig. 4 (stratigraphic constraint, no covariance matrix).


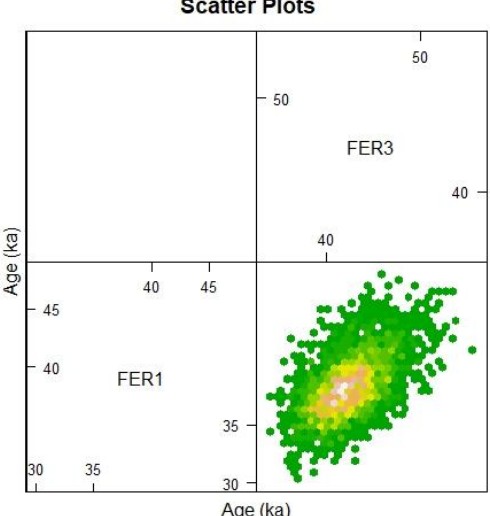

**Fig. 6:** Bivariate scatter plot from the joint posterior distribution of the ages of samples FER 1 and FER
3 when a stratigraphic constraint is applied (sample FER 1 is younger than sample FER 3) and off-
diagonal members of covariance matrix are used to model systematic errors (note: in this case, for
illustrative purposes we used a simplistic covariance matrix – see section 5.3.1. for details). The
truncation in the upper-left hand corner scatter plot indicates the effect of the stratigraphic constraint.

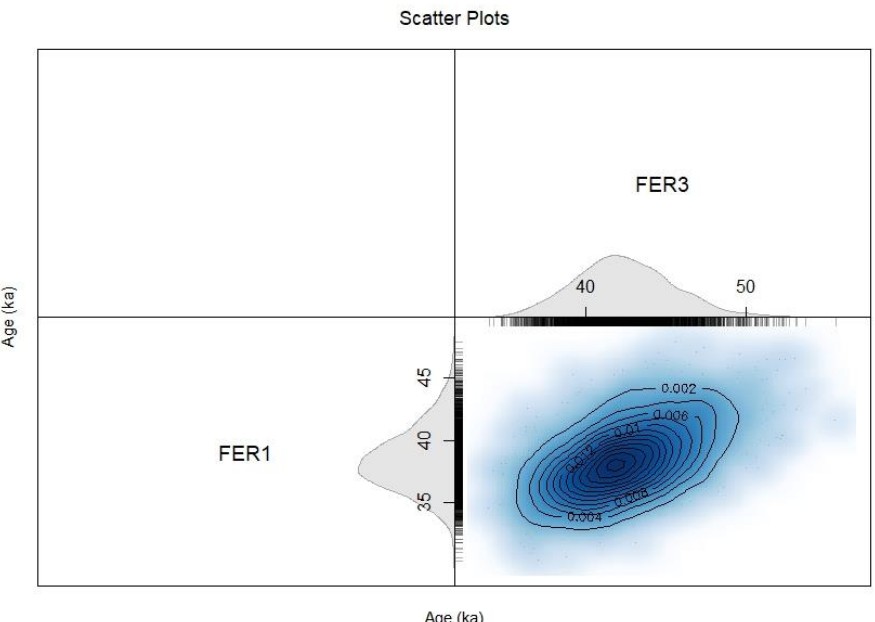

**Fig. 7:** Probability densities for the OSL ages estimated jointly, using the same model as that
implemented to generate Fig. 6 (stratigraphic constraint and off-diagonal members in the covariance
matrix). The positive correlation in the joint posterior density reflects the effect of modelling the
systematic errors with a covariance matrix (and, to some degree, of the stratigraphic constraint).





**Table 1.** Summary of Credible Intervals for the ages (in ka) of samples FER 1 and FER 3 estimated in
the different modelled scenarios.

| Sample | 68% Confidence Interval | | 95% Confidence Interval | |
|---|---|---|---|---|
| | lower | upper | lower | upper |
| Independent | | | | |
| FER 1 | 36.0 | 40.5 | 34.1 | 43.3 |
| FER 3 | 38.9 | 44.6 | 36.6 | 47.8 |
| In stratigraphy | | | | |
| FER 1 | 36.2 | 40.4 | 34.3 | 42.9 |
| FER 3 | 40.0 | 45.0 | 38.1 | 48.5 |
| No stratigraphic constraint, with 'simplistic' covariance (section 5.3.1) | | | | |
| FER 1 | 36.0 | 40.8 | 33.9 | 43.8 |
| FER 3 | 39.2 | 45.4 | 36.7 | 48.1 |
| In stratigraphy, with realistic covariance (section 5.3.2) | | | | |
| FER 1 | 36.1 | 40.5 | 34.2 | 42.6 |
| FER 3 | 39.8 | 45.3 | 37.8 | 48.6 |
| In stratigraphy, with covariance and a 'young' radiocarbon age | | | | |
| FER 1 | 35.2 | 39.4 | 33.3 | 41.2 |
| FER 3 | 39.2 | 42.2 | 36.9 | 42.3 |
| In stratigraphy, with covariance and an 'old' radiocarbon age | | | | |
| FER 1 | 38.7 | 43.5 | 36.2 | 46.2 |
| FER 3 | 46.1 | 48.7 | 46.1 | 51.5 |





**Table 2.** List of physical units and associated uncertainties used in this work. The letter $i$ in subscript
indicates a sample specific value, its absence a common value shared between samples. The letter $s$
indicates absolute uncertainties, while $\sigma$ is used for relative uncertainties.

| Physical unit | Notation | Systematic uncertainty | Random uncertainty |
|---|---|---|---|
| Laboratory source dose rate | $\dot{d}_{lab}$ | $\sigma_{lab}$ | |
| Cosmic dose rate | $\dot{d}_{cosmic,i}$ | | $s_{cosmic,i}$ |
| K concentration | $[K]_i$ | $\sigma_K$ | $\sigma_{K,i}$ |
| U concentration | $[U]_i$ | $\sigma_U$ | $\sigma_{U,i}$ |
| Th concentration | $[Th]_i$ | $\sigma_{Th}$ | $\sigma_{Th,i}$ |
| Internal dose rate | $\dot{d}_{int}$ | $s_{int}$ | |
| Gamma dose rate | $\dot{d}_{\gamma,i}$ | $\sigma_\gamma$ | $\sigma_{\gamma,i}$ |
| Water content | $WF_i$ | | $s_{WF,i}$ |
