# Peer review of "Towards an improvement of OSL age uncertainties: modelling OSL ages with systematic errors,"

_Geochronology, 2020_

## Referee Comment (RC1) · Andrew Millard (Referee) · 5 Jan 2021

General comments

This paper gives an overall description of the BayLum package which runs in the R statistical computing environment. In general it is clear and only minor changes are needed for publication.

The paper gives a thorough description of the models developed which form a holistic approach to the uncertainties in OSL dating, though it is yet to be seen how well the

input routines will adapt to the slightly differing approaches of different laboratories. The error terms for the various measurements and constants that contribute to the calculation of an OSL date are comprehensively considered and each is handled in an appropriate fashion

Specific and technical comments

I have only one correction that needs to be made. The authors often use the frequentist term confidence interval or its abbreviation CI when they should be using the Bayesian term credible interval. The only place that confidence interval is appropriate is in referring to the Gelman-Rubin statistic. Elsewhere it is not appropriate, for example, Line 195, Figure 1, and in Table 1 the caption says credible intervals (correct) but the header row uses confidence intervals (wrong).

I have run all the code provided in the R markdown file, and confirm that it produces the expected results, except at two points:

In the RMd file the scatterplots generated at lines 254 and 260 in Example 2 do not appear to be truncated as in Figures 4 and 5. I think this is because they use A$Sampling rather than A1$Sampling.

Fer 2 has 95% credible interval of 36-46 ka (manuscript line 526) but the markdown output (from line 423) gives 31-39 ka.
* * *

---

## Referee Comment (RC2) · Ed Rhodes (Referee) · 6 Jan 2021

General comments and observations

I think this paper represents a very useful opportunity to re-examine the nature of uncertainty in OSL dating, in particular the degree to which a formalized treatment of systematic errors can lead to significant enhancement of chronological information, and improvement in the quantification of overall age uncertainty. I feel this is a significant step forwards, and there are several aspects I particularly find attractive.

[Figure]

Firstly, I find very appealing the notion of carefully quantifying all the aspects of the data that feed into several OSL age estimates from a single site, and then performing a single analysis that takes account of all of these.

Secondly, absolute clarification of the shared, part shared or unshared aspects of possible error contributions helps researchers carefully consider ways to develop chronologies, for example with collection of further samples, or ways to improve future dating campaigns, besides improving the uncertainties for existing datasets.

The description of the covariance matrix I found particularly lucid, and the formal declarations of the different contributing components form a useful record. The paper evolves around particular samples from the important Palaeolithic site of La Ferrassie measured in Bordeaux at the IRAMAT-CRP2A laboratory, and carefully steps through the introduction of more constraints and application of greater degrees of information (stratigraphic and relating to shared error terms). There is an understandable concentration in the particular aspects of the techniques used in the dating of these samples, but in later sections of the paper, I was made to feel more confident that I could extend this approach to my own data using some different dose rate approaches.

I have taken a quick look at the guidance notes provided as Supplementary Materials, and these look excellent, clear and very full - really quite an impressive degree of documentation and provision of components. I have not tried to run these yet, but I do intend to; I recently (re)installed R (after a break of several years without it) to interpret OSL/IRSL data, so I feel this will not be too great a step.

In summary, this is a very useful addition, both as a specific set of tools, but also as a prompt for readers to consider error and uncertainty contributions in detail. Possibly the use will be limited to a few specific groups at first, but to me this paper clearly points the way forward. Reading the paper was surprised me that I was not more challenged, in terms of understanding, maths or concepts, and I congratulate the authors on their clarity. I have a series of specific points for consideration listed below, but in general, I

find the paper very clear and with a good narrative that develops throughout.

Some specific comments

Line 35 Add "the user" before "to reproduce. . ."

Line 38 "allows the dating of". Note Huntley et al. never refer to "OSL dating" but use the term "optical dating".

Line 42 Replace "important" with "significant"?

Line 49 I think it would be useful to add "at 1 sigma (68% probability)" or similar here, as many geochronological techniques quote their uncertainties at 2 sigma.

Line 50 Note that there are many ways in which a single OSL sample may correctly not be associated with a single age estimate; for examples samples collected across unobserved stratigraphic boundaries. If the "unit of analysis" is to be a single sample (rather than a single depositional event) then some clarification of the assumptions regarding how this relates to depositional events or post- depositional mixing may be important.

Lines 50-51 Why are field observations or measurements e.g. in-situ gamma spectrometer measurements that play a very important role in reducing uncertainties specifically excluded here? Why is it important to define the "system of analysis" as the laboratory. I withhold judgement on the importance of this, but I highlight here the absolute paramount importance of stratigraphic information based on detailed specific field observations both for lithostratigraphic and morphostratigraphic relationships between different sedimentary units and luminescence samples. I note that this aspect is discussed towards the end (lines 651-655).

Lines 52-57 Yes, I agree with this definition. But, I note that random errors in calibration are not the only contributors to systematic error (line 54) but are included. Other contributors to this error can include mistaken application of the (somewhat complex) theoretical conversion between dose deposited in different materials, or application of

attenuation models.

Line 96 Note that there are some instances where radiocarbon dating and luminescence DO share systematic error, for example transport in drilling mud of organic material and sediment grains from one horizon to another during coring; in this situation, both techniques can provide similar erroneous age error of the horizon with material transported into it. This might seem like a rather specialised example, but it is a situation I encounter regularly (at least potentially, meaning I must consider it), and can happen by other processes (e.g. animal burrows, as are common in cave environments). I suggest simply introducing the word "usually" or "typically" after "which".

Line 98 I find the division of uncertainties into just these two categories (systematic and random errors) potentially confusing, and it seems, outdated. The authors already gave an example of where random measurement errors of calibration standards become a systemic error for that laboratory; this can work at many scales, including site or catchment specific errors (for example behaviour of a mineral group in one region affecting the assumptions embedded within the SAR protocol). I note that this type of effect has been discussed within the Bayesian OSL literature for almost two decades, so it seems slightly odd to ignore it here, at least with a comment or reference.

Lines 99-101 Great, this document with application examples is warmly welcomed!

Line 175 I think it would be useful to outline a few characteristics of the Cauchy distribution here for readers, as it is not a term in general circulation within the luminescence dating community, as far as I am aware. I would add that it is also known as the Lorenz distribution, and that it represents a distribution derived from the ratio of two normal (Gaussian) distributions.

Lines 205-208 This is an important and exciting result (the better agreement with independent age estimates). Can the authors tell us what the main reason for this is? I wonder whether it involves the use of KDE (which effectively gives all results a similar weighting) in contrast to the central age or central dose methods that weight the results inversely to their measurement uncertainties. A brief explanation of the observed difference, even if this is just a probabilistic inference would be useful here (e.g. "We suspect that main reason for this difference in combined dating results is. . ..").

Lines 210-212 It is important to say "stratigraphically above", not simply "above", as archaeological contexts are often not flat, and often have significant steep features such as pits dug by people for fires, as middens, or for burials, as well as collapse structures or other "natural" features. The application of the "principle of superposition" being applied here requires certain assumptions to be met, and I think it is worth stating this fact, even if the assumptions themselves are not listed. The current presentation, with no mention of the principle involved, no mention of the assumptions required to be met, and the use of the word "know" is too strong, in my opinion. This is the first introduction of the incorporation of stratigraphic constraints in the paper, and this is one of the most important aspects of Bayesian methods in the application of chronological techniques, so I think it's important to be quite rigorous here. To clarify, I strongly advocate the use of such additional, stratigraphic information in building sediment chronological models, but when introducing readers to these ideas, I also advocate careful attention to detail.

Line 219 Where you say "1 values" this is a little confusing to read, and I misunderstood this initially. I suggest changing it to with "the first row contains the value 1 in each column".

Lines 220-229 Yes, I will need to read the Markdown document to follow what this means; it is a little hard to comprehend simply by reading the text, as acknowledged by the authors' comments. On line 220,I suggest that it is vital to use different terms for stratigraphy and age (in part because they go in opposite directions). Here, you refer to the "lower age bound" meaning "younger age constraint" but this could easily be misunderstood by readers as "the lower stratigraphic constraint on age"; this is confusing as low in age = high in stratigraphy (when not inverted etc), so I suggest reserving the spatial terms "low" and "high" for stratigraphy, and using "younger" and "older" when you mean age.

[Figure]

Line 242 "Should not allow solving" needs changing, as it is incorrect English. But I don't understand what is meant here, so I can't easily suggest alternative wording. Please clarify and reword.

Lines 241-267 This is a very fair and useful introduction to this issue, and many of these issues have clearly been discussed in earlier publications by this research group. I still find the term "systematic errors" as used here somewhat dangerous, as they may be only systematic to this particular group of samples (and therefor probably don't count as true systematic errors). Perhaps I am being too pedantic, but it would be clearer to call these uncertainties "site-systematic" or similar. This is a key issue, and one that it is important for readers to grasp, so I don't want to change the dynamic of the text or narrative flow here.

Line 278 I think the comma after "matrix" is wrongly placed? I think the word "that" could be inserted at this point to help clarify this sentence.

Line 371 I'm not clear what "(resp. Th)" means. Is this "with respect to thorium"?

Line 373 I think this should be "...two systematic sources..." at the end of the line. Line 379 This is probably a good idea to use a single fixed value for quartz, but I think the amount should possibly be varied as a function of the grain size used?

Line 390 This section seems to omit discussion of the obvious solution to this problem that has been widely used for very many decades – the use of a portable NaI or similar calibrated gamma spectrometer. I think this section needs some work to generalize it for a wider audience and focus less specifically on those techniques used at IRAMAT-CRP2A and include discussion of methods that have been developed and applied specifically to reduce the introduction of such systematic errors (e.g. direct measurement of Th230, U238 and K using small samples for beta dose rate determination, for example using ICP-OES/MS or NAA which significantly reduces this sort of error). OK, I see that this discussion appears starting at line 558, so perhaps simply include a reference or pointer to that discussion in this section?

Lines 397-425 This feels like quite a lot of discussion about water content without really discussing some of the important issues. First, although modern values are indeed widely used (and in some cases this may be appropriate), workers often also commonly model past water content, and occasionally water content changes. This is not well represented here it seems to me, and there is little guidance on how to include the different uncertainty aspects involved in that modelling. I note that water content uncertainty represents a good example of a "middle ground" error type; highly sample-systematic, partly site-systematic, and not at all truly systematic (i.e. certainly does not affect every sample to the same degree) and with a different impact on different samples even when the changes are similar.

Lines 427-8 I note that the labelling for equations is somewhat irregular; some equations have numbers, others do not (e.g. all the water content discussion equations).

Line 426 Did I miss it, or was theta defined somewhere?

Line 435 Missing comma after "beta dose rates"

Lines 416-451 I find this description and explanation very clear! This gives me additional confidence and enthusiasm to attempt this approach myself, which is great, I feel.

Lines 569-571 I disagree that counting methods such as beta counting do not have a systematic error that can be quantified and applied. In fact, in my limited experience (i.e. reports from others), G-M beta counting is dominated by systematic not random (i.e. counting) errors, and suites of samples measured using this technique probably have significant group-systematic error introduced that could be subsequently calibrated, and the value included in the covariance matrix.

Lines 577-580 It may be hard to isolate, but there certainly is a very significant systematic error introduced by the threshold technique; in fact, the counting uncertainties are commonly negligible (e.g. many hundreds of thousands of counts even for short measurement with small diameter crystal). However, the conversion to dose rate depends critically on several things – the dose rate value of the sites used in the calibration; the uncertainties in these values are significant for the calibration holes located around the Massif Central, as an example. Secondly, the location of the threshold energy can be tricky, as most NaI systems are quite temperature-dependent, so it is necessary to locate and fit specific emission lines, and in low dose rate sites (such as in limestone caves), these can be hard to locate. Thirdly, damage to the crystal can gradually change the spectrometer sensitivity over time. These effects can be time- or site-systematic, but can readily be quantified by regular re-calibration.

Lines 587-8 I'm not clear what is meant by the phrase "attenuation in grains implies that something other than the grains does not attenuate radiation". Please clarify.

Line 618 Parentheses open but do not close.

Line 622 Replace "equivalents doses" with "equivalent doses".

Lines 665-682 Yes, I firmly agree with these conclusions!

―――――――――――――――――――――

---

## Author Comment (AC1) · 9 Feb 2021

General review comments by Andrew Millard

This paper gives an overall description of the BayLum package which runs in the R statistical computing environment. In general it is clear and only minor changes are needed for publication.

We thank the referee for this positive feedback.

The paper gives a thorough description of the models developed which form a holistic approach to the uncertainties in OSL dating, though it is yet to be seen how well the input routines will adapt to the slightly differing approaches of different laboratories.

We agree with the referee on this issue of laboratory specificities, assumptions and approaches. We would like to take this opportunity to tell interested readers that we would be happy to help adapt the routines to particular cases that we have not considered here.

The error terms for the various measurements and constants that contribute to the calculation of an OSL date are comprehensively considered and each is handled in an appropriate fashion.

Thank you!

Specific and technical comments

I have only one correction that needs to be made. The authors often use the frequentist term confidence interval or its abbreviation CI when they should be using the Bayesian term credible interval. The only place that confidence interval is appropriate is in referring to the Gelman-Rubin statistic. Elsewhere it is not appropriate, for example, Line 195, Figure 1, and in Table 1 the caption says credible intervals (correct) but the header row uses confidence intervals (wrong).

Agreed.

I have run all the code provided in the R markdown file, and confirm that it produces the expected results, except at two points: In the RMd file the scatterplots generated at lines 254 and 260 in Example 2 do not appear to be truncated as in Figures 4 and 5. I think this is because they use A$Sampling rather than A1$Sampling.

Thanks a lot for such a thorough review! Well spotted, we have corrected the Rmd file accordingly.

Fer 2 has 95% credible interval of 36-46 ka (manuscript line 526) but the markdown output (from line 423) gives 31-39 ka.

This is because we have not provided, together with the Rmd file, the BIN-files containing all grains analysed in the paper, as we wrote p. 4, l. 156-157: 'Each folder contains one BIN/BINX-file (i.e. OSL measurements; note that only a small fraction of the measured grains is included here in Supplementary Material)'. In the Rmd-file, l. 85 we also wrote 'Please note that the example datasets have been considerably shortened and do not include the full dataset of the samples FER1 and FER3'.

---

## Author Comment (AC2) · 9 Feb 2021

General review comments by Ed Rhodes

**General comments and observations**

I think this paper represents a very useful opportunity to re-examine the nature of uncertainty in OSL dating, in particular the degree to which a formalized treatment of systematic errors can lead to significant enhancement of chronological information, and improvement in the quantification of overall age uncertainty. I feel this is a significant step forwards, and there are several aspects I particularly find attractive.

We warmly thank the referee for such a positive feedback.

Firstly, I find very appealing the notion of carefully quantifying all the aspects of the data that feed into several OSL age estimates from a single site, and then performing a single analysis that takes account of all of these.

Here we would like to add that we believe, or at least hope that the approach presented here may be useful beyond the scale of a study site. Indeed, chronological questions involving different sites where the same equipment and calibration standards are used may benefit from analyses including all systematic sources of errors.

Secondly, absolute clarification of the shared, part shared or unshared aspects of possible error contributions helps researchers carefully consider ways to develop chronologies, for example with collection of further samples, or ways to improve future dating campaigns, besides improving the uncertainties for existing datasets.

We also believe that BayLum, or other such modelling tools, may indeed lead to changes in our sampling and measurement strategies; we are happy to see that others think that way, too.

The description of the covariance matrix I found particularly lucid, and the formal declarations of the different contributing components form a useful record. The paper evolves around particular samples from the important Palaeolithic site of La Ferrassie measured in Bordeaux at the IRAMAT-CRP2A laboratory, and carefully steps through the introduction of more constraints and application of greater degrees of information (stratigraphic and relating to shared error terms). There is an understandable concentration in the particular aspects of the techniques used in the dating of these samples, but in later sections of the paper, I was made to feel more confident that I could extend this approach to my own data using some different dose rate approaches.

We are certainly happy to see that the referee is confident that he could extend our approach to his data, but here we reiterate our reply to one of the other referee's comments: we would be happy to help interested readers adapt the routines to particular cases that we have not considered in our paper.

I have taken a quick look at the guidance notes provided as Supplementary Materials, and these look excellent, clear and very full - really quite an impressive degree of documentation and provision of components. I have not tried to run these yet, but I do intend to; I recently (re)installed R (after a break of several years without it) to interpret OSL/IRSL data, so I feel this will not be too great a step.

The other referee spotted one mistake in the Rmd file, which we corrected. We hope there is no error left, but were it to be the case, we would be happy to help.

In summary, this is a very useful addition, both as a specific set of tools, but also as a prompt for readers to consider error and uncertainty contributions in detail. Possibly the use will be limited to a few

specific groups at first, but to me this paper clearly points the way forward. Reading the paper was surprised me that I was not more challenged, in terms of understanding, maths or concepts, and I congratulate the authors on their clarity. I have a series of specific points for consideration listed below, but in general, I find the paper very clear and with a good narrative that develops throughout.

Thanks!

Some specific comments

Line 35 Add "the user" before "to reproduce. . ."

Agreed.

Line 38 "allows the dating of". Note Huntley et al. never refer to "OSL dating" but use the term "optical dating".

Agreed.

Line 42 Replace "important" with "significant"?

Agreed.

Line 49 I think it would be useful to add "at 1 sigma (68% probability)" or similar here, as many geochronological techniques quote their uncertainties at 2 sigma.

Agreed.

Line 50 Note that there are many ways in which a single OSL sample may correctly not be associated with a single age estimate; for examples samples collected across unobserved stratigraphic boundaries. If the "unit of analysis" is to be a single sample (rather than a single depositional event) then some clarification of the assumptions regarding how this relates to depositional events or post-depositional mixing may be important.

Agreed – we added the sentence 'we assume that each sample corresponds to a deposition event, and thus to a single age (no post-depositional mixing is considered)'.

Lines 50-51 Why are field observations or measurements e.g. in-situ gamma spectrometer measurements that play a very important role in reducing uncertainties specifically excluded here? Why is it important to define the "system of analysis" as the laboratory.

Agreed – we have added the sentence: 'It should be emphasised here that field equipment is part of what we call the laboratory; this is important for the definition of what we call systematic errors.'

I withhold judgement on the importance of this, but I highlight here the absolute paramount importance of stratigraphic information based on detailed specific field observations both for lithostratigraphic and morphostratigraphic relationships between different sedimentary units and luminescence samples. I note that this aspect is discussed towards the end (lines 651-655).

We now warn the readers explicitly in the introduction (l.98-100): 'It should be noted that we take here the stratigraphic information for granted, but we warn the user against treating such information lightly, as it bears great consequences on the age calculation.' Then, at the beginning of section 4. 'Stratigraphic constraints', we now refer the reader to the paper where detailed stratigraphic information is given for the site of La Ferrassie (Guérin et al., 2015b). We have left the paragraph mentioned by the referee (l. 651-655, now l. 662-663) untouched.

Lines 52-57 Yes, I agree with this definition. But, I note that random errors in calibration are not the only contributors to systematic error (line 54) but are included. Other contributors to this error can include mistaken application of the (somewhat complex) theoretical conversion between dose deposited in different materials, or application of attenuation models.

Agreed – we have added the sentence 'Of course, other sources of errors may exist (for example when using the infinite matrix assumption to calculate grain size attenuation factors, see e.g. Guérin et al., 2012), but in this article we consider only known, quantified sources of errors.'

Line 96 Note that there are some instances where radiocarbon dating and luminescence DO share systematic error, for example transport in drilling mud of organic material and sediment grains from one horizon to another during coring; in this situation, both techniques can provide similar erroneous age error of the horizon with material transported into it. This might seem like a rather specialised example, but it is a situation I encounter regularly (at least potentially, meaning I must consider it), and can happen by other processes (e.g. animal burrows, as are common in cave environments). I suggest simply introducing the word "usually" or "typically" after "which".

Agreed.

Line 98 I find the division of uncertainties into just these two categories (systematic and random errors) potentially confusing, and it seems, outdated. The authors already gave an example of where random measurement errors of calibration standards become a systemic error for that laboratory; this can work at many scales, including site or catchment specific errors (for example behaviour of a mineral group in one region affecting the assumptions embedded within the SAR protocol). I note that this type of effect has been discussed within the Bayesian OSL literature for almost two decades, so it seems slightly odd to ignore it here, at least with a comment or reference.

Agreed – we now refer the reader to Rhodes et al. (2003) for a discussion of errors affecting OSL ages in a Bayesian framework.

Lines 99-101 Great, this document with application examples is warmly welcomed!

Thanks.

Line 175 I think it would be useful to outline a few characteristics of the Cauchy distribution here for readers, as it is not a term in general circulation within the luminescence dating community, as far as I am aware. I would add that it is also known as the Lorenz distribution, and that it represents a distribution derived from the ratio of two normal (Gaussian) distributions.

Agreed, we have added the two following sentences: 'A Cauchy distribution (sometimes also called Lorentz distribution) is a symmetric distribution which was chosen by Combès et al. (2015) because it has heavy tails, *i.e.* extreme values have a non-zero probability. Hence, the Cauchy distribution seemed to be well-suited for the analysis of widely-dispersed datasets including outlier values such as single grain $D_e$ distributions.' This way, we hope to clarify why the Cauchy distribution was used as an option in the first place.

Lines 205-208 This is an important and exciting result (the better agreement with independent age estimates). Can the authors tell us what the main reason for this is? I wonder whether it involves the use of KDE (which effectively gives all results a similar weighting) in contrast to the central age or central dose methods that weight the results inversely to their measurement uncertainties. A brief explanation of the observed difference, even if this is just a probabilistic inference would be useful here (e.g. "We suspect that main reason for this difference in combined dating results is. . ..").

To put it simply, the CDM (or CAM if we use the original terminology) is a biased model which generally leads to age underestimation (Guérin et al., 2015a, c), contrary to models such as the Average Dose Model (Guérin et al., 2017) or BayLum when using either the lognormal-average or the Gaussian model (Philippe et al., 2018; Heydari and Guérin, 2018). This is because the median of the lognormal distribution fitted to the De distribution in the CAM is always smaller than the mean, whereas we determine the average (mean) dose rate. In other words, the KDE (or any other technical tool) is not the cause of underestimation by the CAM; the CAM leads to age underestimates for a merely mathematical reason, i.e. the CAM is ill-suited given what we measure for the dose rate term of the age equation.

We have rephrased the corresponding paragraphs as follows:

'If we assume that this overdispersion arises from dose rate variability to single grains of quartz, then Heydari and Guérin (2018), using laboratory-controlled experiments, showed that the Cauchy distribution and the CDM should both lead to ~5-10% age underestimation, because both models are biased (Guérin et al., 2015a, c, 2017 for theoretical and experimental proofs). […] Indeed, Guérin et al. (2017) formally demonstrated that the median of the lognormal distribution (as used in the CDM) is a biased estimator and leads to age underestimates when dose rates are dispersed.

[…] The OSL and radiocarbon ages are in good agreement, which was not the case when calculating the ages with the CDM (38 ± 2 ka; this OSL age corresponds to ~15 % underestimation and is broadly consistent, within uncertainties, with theoretical predictions stated above). Thus, even without further modelling, the 'BayLum' lognormal-average model seems to provide OSL ages in better agreement with radiocarbon'.

Lines 210-212 It is important to say "stratigraphically above", not simply "above", as archaeological contexts are often not flat, and often have significant steep features such as pits dug by people for fires, as middens, or for burials, as well as collapse structures or other "natural" features. The application of the "principle of superposition" being applied here requires certain assumptions to be met, and I think it is worth stating this fact, even if the assumptions themselves are not listed. The current presentation, with no mention of the principle involved, no mention of the assumptions required to be met, and the use of the word "know" is too strong, in my opinion. This is the first introduction of the incorporation of stratigraphic constraints in the paper, and this is one of the most important aspects of Bayesian methods in the application of chronological techniques, so I think it's important to be quite rigorous here. To clarify, I strongly advocate the use of such additional, stratigraphic information in building sediment chronological models, but when introducing readers to these ideas, I also advocate careful attention to detail.

Agreed. First, we have added the word 'stratigraphically' in front of 'above'. Then, at the end of the sentence considered here, we added the brackets '(for detailed stratigraphic information at the site of La Ferrassie, which is of paramount importance in this section, the reader is referred to Guérin *et al*., 2015b)'. Finally, as already mentioned above as a response to a previous comment, we added the following sentence in the introduction: 'It should be noted that we take here the stratigraphic information for granted, but we warn the user against treating such information lightly, as it bears great consequences on the age calculation.' We hope we have now put enough warnings…

Line 219 Where you say "1 values" this is a little confusing to read, and I misunderstood this initially. I suggest changing it to with "the first row contains the value 1 in each column". Agreed.

Lines 220-229 Yes, I will need to read the Markdown document to follow what this means; it is a little hard to comprehend simply by reading the text, as acknowledged by the authors' comments. On line

220, I suggest that it is vital to use different terms for stratigraphy and age (in part because they go in opposite directions). Here, you refer to the "lower age bound" meaning "younger age constraint" but this could easily be misunderstood by readers as "the lower stratigraphic constraint on age"; this is confusing as low in age = high in stratigraphy (when not inverted etc), so I suggest reserving the spatial terms "low" and "high" for stratigraphy, and using "younger" and "older" when you mean age.

Agreed. We admit in passing that we have tried – several times – to reformulate the part on how to fill the matrix containing the stratigraphic constraints, but have not found a better solution than to simply look at the worked example provided in supplementary information. No matter how simple this matrix may be to conceive (and to fill when one has done it once), spelling it out clearly has proven difficult.

Line 242 "Should not allow solving" needs changing, as it is incorrect English. But I don't understand what is meant here, so I can't easily suggest alternative wording. Please clarify and reword.

We have reworded the sentence as follows: 'In the previous calculations, all the variance is treated as random, whereas common, systematic errors affect all ages in the same direction, although to varying degrees (so systematic errors are unlikely to result in stratigraphic inversions).' We hope this is clearer now.

Lines 241-267 This is a very fair and useful introduction to this issue, and many of these issues have clearly been discussed in earlier publications by this research group. I still find the term "systematic errors" as used here somewhat dangerous, as they may be only systematic to this particular group of samples (and therefor probably don't count as true systematic errors). Perhaps I am being too pedantic, but it would be clearer to call these uncertainties "site-systematic" or similar.

We disagree here with the 'site-systematic' wording, as calibration errors will affect all samples measured subsequently, no matter where the samples come from. Nevertheless, we agree with the referee that the context is of paramount importance when it comes to the meaning of the word systematic. This is why in the introduction we have the paragraph clarifying what our system is: 'Note that in what follows, the unit of analysis is a sediment sample; we assume that each sample corresponds to a deposition event, and thus to a single age (no post-depositional mixing is considered). The system of analysis is the laboratory in which the measurements are performed and includes both the apparatus and associated calibration standards. It should be emphasised here that field equipment is part of what we call the laboratory; this is important for the definition of what we call systematic errors'.

This is a key issue, and one that it is important for readers to grasp, so I don't want to change the dynamic of the text or narrative flow here.

We added the words in bold in the following sentence: 'there are almost as many ways of estimating systematic and random uncertainties as there are (combinations of) ways to determine dose rates; **in any case the notion of systematic error is only valid in a given context, which must always be made explicit**'.

Line 278 I think the comma after "matrix" is wrongly placed? I think the word "that" could be inserted at this point to help clarify this sentence.

Agreed.

Line 371 I'm not clear what "(resp. Th)" means. Is this "with respect to thorium"?

We meant respectively Th (this is now spelled out).

Line 373 I think this should be ". . .two systematic sources. . ." at the end of the line.

Even we got a little confused when re-reading the end of this paragraph. We reworded as follows: 'In contrast to the case of [40]K, the analysis of the high-resolution spectra for these radioactive chains is based on a number of primary gamma rays (whereas there is only one ray for K); more specifically, a weighted mean of the concentrations determined from each ray included in the analysis (after taking interference into account) is calculated to estimate the concentration of U (respectively Th). As a result, the standard deviation of the error on the concentration in U (resp. Th) in the sample comes from two sources: the relative standard deviation on the concentration of the standard corresponds to a systematic error and is denoted $\sigma_\mathrm{U}$ (resp. $\sigma_\mathrm{Th}$); conversely, the other relative standard deviations (arising from the counting of the standards and of the sample) are treated as random and quadratically summed to obtain $\sigma_{\mathrm{U},i}$ (resp. $\sigma_{\mathrm{Th},i}$).'

Line 379 This is probably a good idea to use a single fixed value for quartz, but I think the amount should possibly be varied as a function of the grain size used?

Since most of this internal dose rate comes from alpha radiation, it is only weakly dependent on grain size. But most importantly, we are not prescriptive regarding this value – even within the author list some of us make different assumptions. As a result, and to keep the fluidity of the text, we have decided not to expand on this issue.

Line 390 This section seems to omit discussion of the obvious solution to this problem that has been widely used for very many decades – the use of a portable NaI or similar calibrated gamma spectrometer. I think this section needs some work to generalize it for a wider audience and focus less specifically on those techniques used at IRAMAT-CRP2A and include discussion of methods that have been developed and applied specifically to reduce the introduction of such systematic errors (e.g. direct measurement of Th230, U238 and K using small samples for beta dose rate determination, for example using ICP-OES/MS or NAA which significantly reduces this sort of error). OK, I see that this discussion appears starting at line 558, so perhaps simply include a reference or pointer to that discussion in this section?

Agreed – we now refer the reader to section 7.1.

Lines 397-425 This feels like quite a lot of discussion about water content without really discussing some of the important issues. First, although modern values are indeed widely used (and in some cases this may be appropriate), workers often also commonly model past water content, and occasionally water content changes. This is not well represented here it seems to me, and there is little guidance on how to include the different uncertainty aspects involved in that modelling. I note that water content uncertainty represents a good example of a "middle ground" error type; highly sample systematic, partly site-systematic, and not at all truly systematic (i.e. certainly does not affect every sample to the same degree) and with a different impact on different samples even when the changes are similar.

Agreed – we now refer the reader to an article dedicated to the question of water content determination: 'The determination of the water content in the sediment over time is a very difficult task as it involves many different parameters, including past rainfall – see for example Nelson and Rittenour (2015) for a discussion on how to determine water contents depending on sediment grain size, hydrometric regimes, etc.' Also, we now conclude this paragraph as follows: 'Finally, it should be emphasized that uncertainty on water content may well correspond to errors which are neither really random nor really systematic; in our view different modelling choices may be put forward and implemented, depending on the particular sedimentological and pedological context.'

Lines 427-8 I note that the labelling for equations is somewhat irregular; some equations have numbers, others do not (e.g. all the water content discussion equations).

We chose to label only the equations we refer to in the main text.

Line 426 Did I miss it, or was theta defined somewhere?

It was defined in Eq. (1), section 5.1. This is spelled out in the text l. 428 (now l. 451): 'With these considerations in mind on errors and their nature, the corresponding θ matrix (Eq. 1)…'.

Line 435 Missing comma after "beta dose rates".

Agreed.

Lines 416-451 I find this description and explanation very clear! This gives me additional confidence and enthusiasm to attempt this approach myself, which is great, I feel.

Thanks!

Lines 569-571 I disagree that counting methods such as beta counting do not have a systematic error that can be quantified and applied. In fact, in my limited experience (i.e. reports from others), G-M beta counting is dominated by systematic not random (i.e. counting) errors, and suites of samples measured using this technique probably have significant group-systematic error introduced that could be subsequently calibrated, and the value included in the covariance matrix.

Here, we only partly agree with the referee. First, we agree that counting uncertainties (which are random) are really small when using counting techniques. Indeed, most of the errors associated with counting techniques come from the conversion from count rates to dose rates; the problem is that this conversion depends on the radioelement concentrations. As a result, it may well be that a series of samples coming from the same site will have similar concentrations and thus a similar error in the conversation from count rates to dose rates, but to quantify this error and refine the conversion factor, the user must use a method to determine concentrations in K, U and Th. And if such a method is available, then this latter method is probably more precise than counting techniques (than beta counting at least), making the counting technique (virtually) useless.

We have changed our text accordingly (our additions in bold): 'This dependency is a source of error that may not **easily** be characterised by a systematic error (so there is no contribution to the dose covariance matrix)**; indeed, this error on the conversion factor will vary from one site to another depending on the concentrations in K, U and Th (which are generally unknown if a counting technique is used), and even within one site from one sample to another (again by unknown amounts since the variability in K, U and Th is unknown).**'

Lines 577-580 It may be hard to isolate, but there certainly is a very significant systematic error introduced by the threshold technique; in fact, the counting uncertainties are commonly negligible (e.g. many hundreds of thousands of counts even for short measurement with small diameter crystal). However, the conversion to dose rate depends critically on several things – the dose rate value of the sites used in the calibration (really? Miallier et al. (2009) did not observe such a trend as a function of dose rate, contrary to the trend seen with Al2O3 dosimeters on the same sites – Kreutzer et al., 2018); the uncertainties in these values are significant for the calibration holes located around the Massif Central, as an example (but the fitting line is really close to perfect in Miallier et al. (2009)). Secondly, the location of the threshold energy can be tricky, as most NaI systems are quite temperature-dependent, so it is necessary to locate and fit specific emission lines, and in low dose rate sites (such as in limestone caves), these can be hard to locate.

We agree that in situ measurement systems are temperature dependent to some extent. However, this should not be a real practical issue. Newer, recommended, LaBr systems are self-stabilising. NaI systems, if applied properly should be given a chance acclimate and energy calibrated before being used each time in the field (which is a matter of minutes; but it might take hours if temperature differences are large). In low dose-rate sites, such energy calibration can be done using, e.g., using a 'stone' with natural minerals with sufficient abundance of U and Th. Simple KCl would do as well to at least to fit the K-40 gamma line. Additional, minor energy shifts that still occur during the measurement should be negligible.

Besides, we now refer the reader to an article published recently by Lebrun et al. (2020). This paper presents an R package called 'gamma' which locates specific emission lines to perform the energy calibration of the spectra. Then, the threshold technique may be used to convert spectra to dose rates.

Thirdly, damage to the crystal can gradually change the spectrometer sensitivity over time. These effects can be time- or site-systematic, but can readily be quantified by regular re-calibration.

Agreed. To improve our text, we have added the following lines: 'Difficulties in implementing the threshold technique may occur in low dose rates environments, because energy calibration may not be straightforward; in such cases, routines such as those implemented in the R package 'gamma' (Lebrun et al., 2020) may be put to profit. Finally, ageing of the crystal may also result in time-dependent errors – the latter must be taken care of by regular calibration experiments.'

Lines 587-8 I'm not clear what is meant by the phrase "attenuation in grains implies that something other than the grains does not attenuate radiation". Please clarify.

We have added the words in bold: 'attenuation in grains implies that something other than the grains does not attenuate radiation **(i.e. conservation of energy implies that if there is lower dose rate inside a quartz grain, there must be a higher dose rate somewhere else – cf. Guérin *et al.*, 2012a)**'. We hope this is clear enough.

Line 618 Parentheses open but do not close. Agreed.

Line 622 Replace "equivalents doses" with "equivalent doses". Agreed.

Lines 665-682 Yes, I firmly agree with these conclusions! Great – thanks!

---

## Author Response (AR1)

**Associate Editor Decision: Publish subject to minor revisions (further review by editor)** (24 Feb 2021) by Richard Staff

Comments to the Author:

"Towards an improvement of OSL age uncertainties: modelling OSL ages with systematic errors, stratigraphic constraints and radiocarbon ages using the R package 'BayLum'" (gchron-2020-40) by Guérin et al. provides a well written, detailed discussion of a robust statistical approach to dealing with uncertainty in luminescence dating.

The two reviews are very thorough, and overwhelmingly positive towards the manuscript (e.g., "…this is a significant step forwards, and there are several aspects I particularly find attractive…"), and the authors' responses to the comments/suggestions of the reviewers are similarly thorough.

I therefore recommend that the paper be accepted, pending incorporation of these revisions.

I have only a couple of additional, extremely minor (wording) comments myself:

* In the first line of the Abstract, I would be inclined to spell out "OSL" upon first usage (as you already have in the first line of the Introduction). (I realise that the abbreviation is VERY well known and understood, and so do not propose that the authors also introduce it to the Title, which might be too cumbersome. Nevertheless, for the less specialist reader – who might, sadly(!), only read the Abstract – I suggest the inclusion for completeness.) Agreed.
* L24: I suggest insertion of "…the R package 'BayLum', [introduced/described/presented herein,] allows…" (or similar wording). Agreed.
* L35: change "…allowing to reproduce…" to "…allowing the reproduction of…". Agreed.

* Per the final suggestion on page 4, and your first response on page 5 to the comments from Reviewer 2, I would also think that you mean "with respect to thorium", rather than "respectively Th"? The same is true later in the inserted paragraph where you give, e.g., "resp. $\sigma$Th" where, again, I would think that you mean "with respect to $\sigma$Th"?
I replaced 'respectively Th' by 'and that of Th'. I also replaced 'resp.' everywhere.
* I understand your response to the first comment of Reviewer 2 on page 7 (of your responses). However, I personally would agree with the Reviewer that I think that it would make sense to number all equations in the manuscript, whether referred to herein or not. Part of the logic of this suggestion is that subsequent authors (including in PhD theses) may wish to refer to your work and potentially wish to cite these currently un-numbered equations.
Agreed – all equations are now numbered.
Finally, may I thank the authors, once again, for their interesting and detailed manuscript, which will provide a worthy addition to "Geochronology". I wish them well for their on-going research.
Thank you!

Non-public comments to the author:

Although I have had to categorise the decision as "minor corrections", all this means is the minor corrections that you have already noted in your responses to reviewers

Thanks once again for your very interesting manuscript.

With very best wishes,

Richard